# STEP: Warm-Started Visuomotor Policies with Spatiotemporal Consistency Prediction

**Jinhao Li** [1 2 *]  **Yuxuan Cong** [1 *]  **Yingqiao Wang** [1]  **Hao Xia** [1]  **Shan Huang** [1 2]  **Yijia Zhang** [1]  **Ningyi Xu** [1]
**Guohao Dai** [1 2 3]

## Abstract

Diffusion policies have recently emerged as a powerful paradigm for visuomotor control in robotic manipulation due to their ability to model the distribution of action sequences and capture multimodality. However, iterative denoising leads to substantial inference latency, limiting control frequency in real-time closed-loop systems. Existing acceleration methods either reduce sampling steps, bypass diffusion through direct prediction, or reuse past actions, but often struggle to jointly preserve action quality and achieve consistently low latency. In this work, we propose **STEP**, a lightweight spatiotemporal consistency prediction mechanism to construct high-quality warm-start actions that are both distributionally close to the target action and temporally consistent, without compromising the generative capability of the original diffusion policy. Then, we propose a velocity-aware perturbation injection mechanism that adaptively modulates actuation excitation based on temporal action variation to prevent execution stall especially for real-world tasks. We further provide a theoretical analysis showing that the proposed prediction induces a locally contractive mapping, ensuring convergence of action errors during diffusion refinement. Extensive evaluations on nine simulated benchmarks and two real-world tasks show that STEP with 2 steps can achieve an average 21.6% and 27.5% higher success rate than BRIDGER and DDIM on the RoboMimic benchmark and real-world tasks, respectively. The code is available at https://github.com/Kimho666/STEP.

*Equal contribution [1]Shanghai Jiao Tong University, Shanghai, China [2]Shanghai Innovation Institute, Shanghai, China [3]Infinigence-AI, Shanghai, China. Correspondence to: Guohao Dai <daiguohao@sjtu.edu.cn>.

*Proceedings of the 43rd International Conference on Machine Learning*, Seoul, South Korea. PMLR 306, 2026. Copyright 2026 by the author(s).

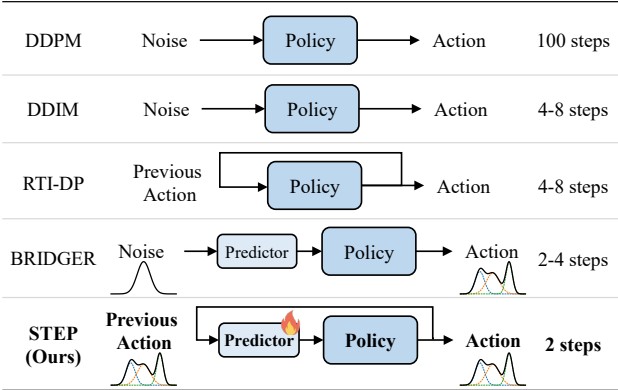

*Figure 1.* Comparison of inference pipelines for diffusion-based visuomotor policies. Our method leverages spatiotemporally consistent action prediction to reduce sampling to only two steps.

## 1. Introduction

Diffusion policy has been introduced as a new paradigm for visuomotor control in robotic manipulation recently (Chi et al., 2025; Ma et al., 2024; Ze et al., 2024; Liu et al., 2025; Wolf et al., 2025; Song et al., 2025). Unlike conventional regression-based policies that directly predict single-step actions, diffusion policies explicitly model the generative distribution of action sequences, formulating action generation as a progressive denoising process that evolves from Gaussian noise toward the target action distribution. This generative formulation naturally captures the multimodality and long-horizon dependencies commonly observed in complex tasks, enabling diffusion-based policies to achieve superior action stability and higher success rates.

However, diffusion policies typically require up to 100 steps to generate high-quality actions, limiting the real-time closed-loop control (Ho et al., 2020). Prior efforts to accelerate diffusion-based action generation mainly fall into three categories. (1) A line of work focuses on accelerating diffusion sampling through improved numerical solvers, including DDIM (Song et al., 2020), DPM-Solver (Lu et al., 2022), and DPM-Solver++ (Lu et al., 2025). These methods reformulate the reverse stochastic diffusion process as a deterministic or higher-order ODE and employ advanced integration schemes to reduce sampling steps. Nevertheless, they still rely on repeated evaluations

of the same policy network, which limits their efficiency in high-dimensional or high-frequency control scenarios. (2) Direct prediction approaches, including CP (Prasad et al., 2024), OneDP (Wang et al., 2025), BRIDGER (Chen et al., 2024), and NO-Diffusion (Aizu et al., 2025), aim to reduce diffusion overhead by replacing iterative denoising with a small number of direct prediction or consistency-based refinement steps. These methods learn to map simple latent or noisy distributions toward target action distributions, but the limited capacity of lightweight predictors makes such distribution transformations challenging, often leading to degraded action quality or reduced robustness. (3) Action reuse strategies such as RTI-DP (Duan et al., 2025), SDP (Høeg et al., 2025), RNR-DP (Chen et al., 2025b) and Falcon (Chen et al., 2025a) exploit temporal continuity to warm-start the diffusion process, reducing iterations at inference time, yet their effectiveness relies on smooth dynamics and cannot reliably guarantee action quality under rapid state changes. Although existing methods can reduce inference cost to some extents, they are still insufficient to fully meet the requirements of high-frequency, closed-loop control on resource-constrained platforms.

Motivated by the above limitations, we investigate how to construct a high-quality warm-start initialization for diffusion-based policies that is both distributionally close to the target action manifold and spatiotemporally consistent across successive control steps, while fully preserving the generative expressiveness of the original diffusion policy. To this end, we propose **STEP**, a diffusion-based control framework equipped with a *spatiotemporal consistency prediction* mechanism that initializes the diffusion process in the vicinity of the target action distribution, enabling robust and low-latency control under rapidly changing system states. Rather than replacing or distilling the diffusion model, our approach serves as a principled warm-start strategy that substantially reduces the required number of denoising iterations without compromising action quality or stability. Our main contributions are summarized as follows:

**(1) Spatiotemporal Consistency Prediction Mechanism.** We propose a lightweight spatiotemporal consistency prediction mechanism that generates high-quality warm-start actions aligned with both the target action distribution and temporal dynamics, enabling low-latency diffusion-based control. Our STEP with 2 steps can achieve an average 21.6% and 48.8% higher success rate over spatial-only method BRIDGER (Chen et al., 2024) and temporal-only Falcon (Chen et al., 2025a) method.

**(2) Velocity-aware Perturbation Injection Mechanism.** We propose a lightweight velocity-aware perturbation injection mechanism that prevents execution deadlock especially for real-world robotic deployment by introducing bounded actuation excitation only when necessary, without degrad-

ing control precision. Under different denoising steps, our method can reduce the average episode execution time by 59%, resulting in faster task completion.

**(3) Local Contractivity Analysis.** We provide a theoretical analysis showing that our method can induce a locally contractive mapping, which guarantees convergence of action errors during subsequent diffusion refinement.

We conduct extensive evaluations on nine simulated benchmarks and two real-world robotic tasks. Notably, on the RoboMimic benchmark, STEP with 2 steps can achieve an average 21.6% improvement in task success rate over the state-of-the-art BRIDGER (Chen et al., 2024) method. On real-world tasks, STEP with 2 steps improves average 27.5% in success rate compared to DDIM. These results demonstrate that STEP consistently advances the Pareto frontier of inference latency and success rate over existing methods.

## 2. Background

Diffusion policy models continuous action generation as a conditional diffusion process, enabling expressive representation of complex and multimodal action distributions for robot control. Given an observation $\mathbf{o}$, such as visual inputs or proprioceptive states, the policy generates an action sequence by iteratively denoising a latent action variable conditioned on $\mathbf{o}$.

Let $\mathbf{a}_0 \in \mathbb{R}^{T \times d}$ denote a ground-truth action sequence with horizon $T$ and action dimension $d$. The forward diffusion process gradually corrupts $\mathbf{a}_0$ through a Markov chain:

$$q(\mathbf{a}_t \mid \mathbf{a}_{t-1}) = \mathcal{N}\left(\sqrt{1 - \beta_t}\,\mathbf{a}_{t-1},\, \beta_t \mathbf{I}\right), \qquad (1)$$

where $\{\beta_t\}_{t=1}^N$ defines a noise schedule. This process admits a closed-form reparameterization:

$$\mathbf{a}_t = \sqrt{\bar{\alpha}_t}\,\mathbf{a}_0 + \sqrt{1 - \bar{\alpha}_t}\,\boldsymbol{\epsilon}, \quad \boldsymbol{\epsilon} \sim \mathcal{N}(\mathbf{0}, \mathbf{I}) \qquad (2)$$

with $\alpha_t = 1 - \beta_t$ and $\bar{\alpha}_t = \prod_{i=1}^t \alpha_i$.

The policy learns a conditional reverse process that progressively removes noise given the observation $\mathbf{o}$:

$$p_\theta(\mathbf{a}_{t-1} \mid \mathbf{a}_t, \mathbf{o}) = \mathcal{N}\left(\boldsymbol{\mu}_\theta(\mathbf{a}_t, t, \mathbf{o}), \boldsymbol{\Sigma}_t\right), \qquad (3)$$

where a neural network $\boldsymbol{\epsilon}_\theta(\mathbf{a}_t, t, \mathbf{o})$ is trained to predict the injected noise. Under this parameterization, the mean of the reverse transition is given by:

$$\boldsymbol{\mu}_\theta = \frac{1}{\sqrt{\alpha_t}}\left(\mathbf{a}_t - \frac{1 - \alpha_t}{\sqrt{1 - \bar{\alpha}_t}}\boldsymbol{\epsilon}_\theta(\mathbf{a}_t, t, \mathbf{o})\right). \qquad (4)$$

Training is performed by minimizing a conditional noise prediction loss $\mathcal{L}$:

$$\mathcal{L} = \mathbb{E}_{\mathbf{a}_0, \boldsymbol{\epsilon}, t}\left\|\boldsymbol{\epsilon} - \boldsymbol{\epsilon}_\theta(\sqrt{\bar{\alpha}_t}\mathbf{a}_0 + \sqrt{1 - \bar{\alpha}_t}\boldsymbol{\epsilon}, t, \mathbf{o})\right\|_2^2, \quad (5)$$

which corresponds to optimizing a variational lower bound on the conditional data likelihood.

At inference time, action generation starts from Gaussian noise $\epsilon \sim \mathcal{N}(\mathbf{0}, \mathbf{I})$ and applies the learned reverse process for $N$ denoising steps:

$$\mathbf{a}_{k-1} = \frac{1}{\sqrt{\alpha_k}} \Big( \mathbf{a}_k - \frac{1 - \alpha_k}{\sqrt{1 - \bar{\alpha}_k}} \epsilon_\theta(\mathbf{a}_k, k, \mathbf{o}) \Big) + \sigma_k \epsilon, \quad (6)$$

where $\alpha_k$ and $\bar{\alpha}_k = \prod_{i=1}^k \alpha_i$ are diffusion schedule parameters, $\sigma_k$ controls the injected noise, and $\epsilon_\theta$ is the learned denoising network. Obtaining high-quality action sequences typically requires nearly 100 denoising steps, which introduces substantial inference latency and makes it challenging to deploy diffusion policies in high-frequency, real-time closed-loop control.

## 3. Method

### 3.1. Motivation

Previous methods (Song et al., 2020; Lu et al., 2022; 2025; Prasad et al., 2024; Chen et al., 2024; Duan et al., 2025; Chen et al., 2025b) attempt to accelerate diffusion policies by modifying the initialization or sampling process. To more clearly understand the differences among these approaches, we introduce three forms of consistency for diffusion policies: temporal consistency, spatial consistency and spatiotemporal consistency.

**Temporal Consistency.** Temporal consistency captures the smoothness of actions across successive control steps, reflecting physical continuity and control frequency constraints. Given two consecutive time steps $t-1$ and $t$, with the executed action $a_{t-1}$ and a warm-start action $\tilde{a}_t$ at time $t$, we say the warm start satisfies temporal consistency if

$$\|\tilde{a}_t - a_{t-1}\| \leq \epsilon_t, \quad \forall t, \quad (7)$$

where $\epsilon_t$ is a Lipschitz-like bounded constant determined by the system dynamics and control frequency.

This assumption is commonly exploited by action extrapolation or reuse strategies, which initialize the diffusion process using previous actions. While effective in reducing inter-step action variation, temporal consistency alone does not guarantee that the initialized action remains compatible with the current system state.

**Spatial Consistency.** Spatial consistency characterizes the alignment between the warm-start action and the state-conditioned target action distribution induced by the diffusion policy. Let $s_t$ denote the current system state and $p(a \mid s_t)$ the corresponding target action distribution. A warm-start action $\tilde{a}_t$ satisfies spatial consistency if

$$\mathrm{dist}\big(\tilde{a}_t, \mathcal{M}(s_t)\big) \leq \epsilon_s, \quad \forall t \quad (8)$$

*Table 1.* Comparison of diffusion-based control acceleration methods from the perspective of spatiotemporal consistency.

| Method | TC | SC |
|---|---|---|
| Vanilla DDPM | ✗ | ✗ |
| DDIM/DPM-Solver/DPM-Solver++ | ✗ | ✗ |
| RTI-DP/SDP/RNR-DP/Falcon | ✓ | ✗ |
| CP/OneDP | ✗ | ✗ |
| BRIDGER | ✗ | ✓ |
| **STEP (Ours)** | ✓ | ✓ |

where $\mathcal{M}(s_t)$ denotes the high-probability action manifold of $p(a \mid s_t)$, and $\epsilon_s$ controls the allowable deviation. Methods based on prediction can be viewed as enforcing spatial consistency. However, the predictor in these approaches often leads to limited capability, sacrificing generative flexibility and robustness.

Previous methods can be categorized by the type of consistency they enforce, as summarized in Table 1. Standard diffusion samplers (e.g., DDIM (Song et al., 2020), DPM-Solver (Lu et al., 2022), and DPM-Solver++ (Lu et al., 2025)) as well as distillation-based methods such as CP (Prasad et al., 2024) and OneDP (Wang et al., 2025) generate action sequences from Gaussian noise without warm-starting. As a consequence, neither spatial consistency nor temporal consistency is explicitly enforced across control steps. Methods that reuse historical actions such as RTI-DP (Duan et al., 2025), RNR-DP (Chen et al., 2025b) and Falcon (Chen et al., 2025a) promote temporal consistency through warm-start initialization, yet their initializations are not guaranteed to remain close to the state-conditioned high-probability action manifold, making it challenging to ensure spatial consistency. BRIDGER (Chen et al., 2024) constructs deterministic mappings between noise and actions for faster sampling, but lack explicit constraints on temporal consistency. Our STEP explicitly enforces both spatial and temporal consistency during warm-start initialization, thereby achieving **spatiotemporal consistency** and enabling fast and stable diffusion-based control. A warm-start action $\tilde{a}_t$ is said to be spatiotemporally consistent if it simultaneously satisfies the temporal and spatial consistency in Equation (7) and Equation (8). Spatiotemporal consistency ensures that the warm-start action is both temporally smooth and spatially aligned with the current state-conditioned action distribution, providing a stable and informative initialization for subsequent diffusion refinement.

**Why Spatiotemporal Consistency Matters.** To clarify why enforcing only a single dimension of consistency is insufficient, we analyze two representative failure cases corresponding to existing acceleration strategies.

*Case I: Temporal warm-start without spatial consistency.* Methods such as RTI-DP and Falcon accelerate inference by warm-starting the diffusion process from previous actions, thereby promoting temporal consistency across control steps.

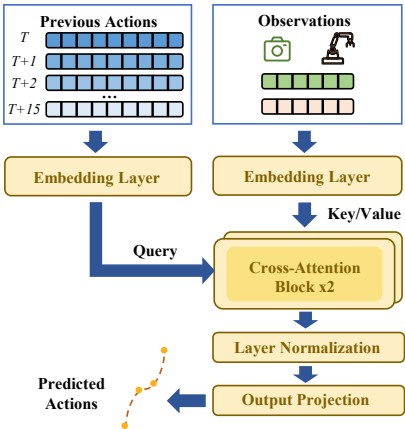

*Figure 2.* The model architecture of predictor in STEP.

However, these warm-start actions are not explicitly constrained to align with the state-conditioned target action distribution. As a result, the initialization may deviate from the high-probability action manifold induced by the current observation, and even with warm-starting, typically still requires additional steps to correct the distributional mismatch.

*Case II: Spatial warm-start without temporal consistency.* Methods like BRIDGER train a conditional predictor to predict actions conditioned on the current state, and use the decoder output as a warm-start initialization for diffusion sampling. While it helps guide the sampling process toward the target action distribution, the predictor operates in a single-shot manner with Gaussian noise as input and does not explicitly condition on previous actions or control steps. As a result, the warm-start initialization varies across consecutive timesteps $t$, making it difficult to guarantee the temporal consistency and potentially leading to oscillatory behaviors. Moreover, since the predictor provides only a coarse initialization rather than directly modeling the local action mode, additional denoising steps are often required to refine the action.

*Table 2.* MARE comparison of SC, TC and STC (SpatioTemporal Consistency).

| Method | Push-T | Lift | Can | Square | Trans. | ToolH. | avg. |
|---|---|---|---|---|---|---|---|
| SC | 0.042 | 0.061 | 0.551 | 0.429 | 0.042 | 0.914 | 0.116 |
| TC | 0.022 | 0.047 | 0.103 | 0.263 | 0.022 | 0.384 | 0.140 |
| STC | 0.023 | 0.041 | 0.104 | 0.052 | 0.023 | 0.051 | **0.049** |

We compute the mean action relative error (MARE) to present two critical issues in Table 2:

- **SC**: Modeling from noise to target action is weak, leading to large discrepancies between predicted actions and ground-truth actions.

- **TC**: Historical actions are not equivalent to target actions, as consecutive action sequences exhibit substantial differences.

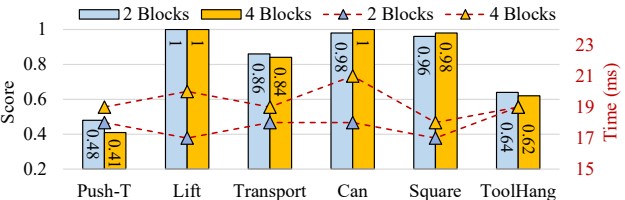

*Figure 3.* Score and latency with different numbers of cross-attention blocks.

Rather than directly predicting target actions from scratch (BRIDGER) or naively reusing historical actions (RTI-DP), the spatiotemporal prediction operates in a residual action space, which lies in a low-entropy, Lipschitz-bounded distribution due to the temporal smoothness.

### 3.2. Spatiotemporal Consistency Prediction

We consider a discrete-time visuomotor control problem with a planning horizon $H$. At each timestep $t$, the policy receives an observation $\mathbf{o}_t \in \mathcal{O}$ and generates a continuous action sequence:

$$\mathbf{A}_t = (a_t, a_{t+1}, \ldots, a_{t+H-1}) \in \mathcal{A}^H, \qquad (9)$$

To exploit spatiotemporal consistency, we define a predictor:

$$f_\theta : \mathcal{O} \times \mathcal{A}^H \to \mathcal{A}^H, \qquad (10)$$

which maps the current observation $\mathbf{o}_t$ and the previous timestep's action sequence $\mathbf{A}_{t-H}$ to a prediction:

$$\hat{\mathbf{A}}_t = f_\theta(\mathbf{o}_t, \mathbf{A}_{t-H}). \qquad (11)$$

The predictor is implemented as a multi-layer Transformer with cross-attention, as shown in Figure 2. Historical actions and current observations are first projected to a shared 128-dimensional embedding space, followed by cross-attention to incorporate temporal context. The predictor is trained in a supervised manner using MSE loss:

$$\mathcal{L}_{\text{pred}}(\theta) = \mathbb{E}\big[\|\hat{\mathbf{A}}_t - \mathbf{A}_t\|_2^2\big]. \qquad (12)$$

This learning objective encourages $\hat{\mathbf{A}}_t$ to approximate the conditional expectation $\mathbb{E}[\mathbf{A}_t \mid \mathbf{o}_t, \mathbf{A}_{t-H}]$. After separate training of the predictor and diffusion policy, they are cascaded for inference.

During inference, the predicted sequence $\hat{\mathbf{A}}_t$ is used to initialize the diffusion reverse process at an intermediate step $K' < K$ rather than from pure noise:

$$\tilde{\mathbf{A}}_{K'} = \sigma \hat{\mathbf{A}}_t + \sigma_t \, \boldsymbol{\epsilon}_t, \quad \boldsymbol{\epsilon}_t \sim \mathcal{N}(\mathbf{0}, \mathbf{I}), \qquad (13)$$

where $\sigma$ and $\sigma_t$ controls the predicted action and noise magnitude which are discussed in 3.3. The reverse denoising process then proceeds from step $K'$ to 0, producing the final

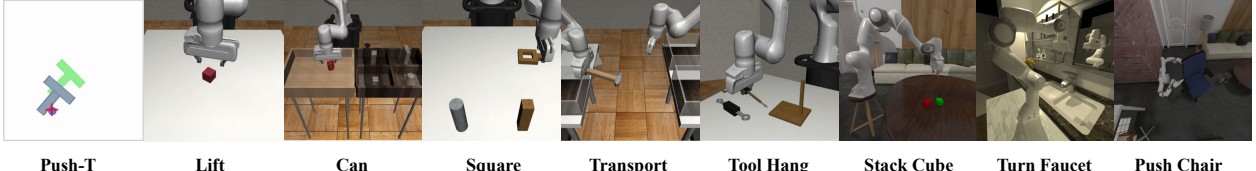

*Figure 4.* Illustrations of Simulation Experiments. Left to Right: Push-T, RoboMimic's 5 tasks, and ManiSkill2's 3 tasks.

---

**Algorithm 1** Warm-Started Diffusion Policy Inference

---

1: **Input:** $K$ (total diffusion steps), $K' < K$ (warm-start step), $\sigma_t$ (noise magnitude)
2: **Initialize:** cache for previous H-step actions, $\mathbf{A}_{\text{cache}} \leftarrow$ None
3: **while** control loop is running **do**
4:     Observe current state $\mathbf{o}_t$
5:     **if** $t < H$ **or** $\mathbf{A}_{\text{cache}} =$ None **then**
6:         Initialize $\mathbf{A}_K \sim \mathcal{N}(\mathbf{0}, \mathbf{I})$
7:     **else**
8:         Predict $\hat{\mathbf{A}}_t = f_\theta(\mathbf{o}_t, \mathbf{A}_{\text{cache}})$
9:         Warm-start: $\tilde{\mathbf{A}}_{K'} = \sigma \hat{\mathbf{A}}_t + \sigma_t \, \boldsymbol{\epsilon}_t, \quad \boldsymbol{\epsilon}_t \sim \mathcal{N}(\mathbf{0}, \mathbf{I})$
10:     **end if**
11:     Run reverse process from step $K'$ to 0 to obtain $\mathbf{A}_t$
12:     Update cache: $\mathbf{A}_{\text{cache}} \leftarrow \mathbf{A}_t$
13:     Execute actions in $\mathbf{A}_t$
14: **end while**

---

action sequence $\mathbf{A}_t$. The entire sequence $\mathbf{A}_t$ is executed and stored in a cache for the next timestep's predictor input.

In Figure 2, the previous action sequence and observations are first encoded by embedding layers, followed by multiple cross-attention blocks (Vaswani et al., 2017). The action sequence serves as the query, while the observations are used as the key and value. The attention outputs are then transformed via layer normalization and an output projection to match the dimensionality of the input action sequence. As shown in Figure 3, experiments on the RoboMimic benchmark indicate that 2 blocks can achieve the same success rate (score) with lower inference latency. Consequently, we adopt two blocks in all subsequent experiments.

### 3.3. Velocity-aware Perturbation Injection

In real-world execution, we observe that the robot may enter an execution deadlock where consecutive actions exhibit minimal variation, leading to insufficient actuation to overcome static friction and control dead zones. To detect such stagnation, we measure the action variation between consecutive timesteps $(..., t - 2H, t - H, t, ...)$ as $\Delta \mathbf{A}_t = \mathbf{A}_{\text{cache}} - \mathbf{A}_{t-2H}$. Execution stagnation is identified using a threshold on the action variation: $\mathbb{I}_t = \mathbb{I}(\|\Delta \mathbf{A}_t\| < \epsilon_a)$, where $\mathbb{I}(\cdot)$ denotes the indicator function and $\epsilon_a$ is a small constant.

Based on this criterion, we switch between normal execution and perturbed execution by adjusting both the action scaling factor and the perturbation magnitude. Then, the warm-

started action is defined as Line 9 in Algorithm 1, where the coefficients are set as

$$\sigma = \begin{cases} \sigma_{\text{scale}}, & \mathbb{I}_t = 1, \\ 1, & \mathbb{I}_t = 0, \end{cases} \quad \sigma_t = \begin{cases} \sigma_{\text{stall}}, & \mathbb{I}_t = 1, \\ 0, & \mathbb{I}_t = 0, \end{cases} \quad (14)$$

where $\sigma_{\text{scale}}$ and $\sigma_{\text{stall}}$ are hyperparameters to controls the predicted action and noise magnitude. This formulation injects controlled actuation excitation only when execution stagnation is detected, while preserving the original policy behavior during normal execution. In practice, we set $\epsilon_a = 0.01$. A small perturbation $\sigma_{\text{stall}} = 0.1$ is sufficient in simulation, whereas real-world experiments require larger $\sigma_{\text{stall}}$ to compensate for static friction and actuation dead zones in Appendix B.

### 3.4. Local Contractivity Analysis

We analyze the local contractive behavior of diffusion-based policy solvers when initialized from a spatiotemporally consistent warm start. Our analysis applies uniformly to commonly used reverse solvers.

**Unified reverse update.** All solvers considered in this work admit a unified reverse update of the form

$$\mathbf{A}_{k-1} = \mu_k(\mathbf{A}_k, \mathbf{o}_t) + \boldsymbol{\xi}_k, \quad (15)$$

where $\mu_k$ denotes the reverse posterior mean induced by the learned denoising network, and $\boldsymbol{\xi}_k$ is a (possibly zero) noise term whose variance is determined by the scheduler. DDPM, DDIM, and DPM-Solver differ in the exact form of $\mu_k$ and $\boldsymbol{\xi}_k$, but all share the same mean-based update structure.

We focus on the contraction property of the mean mapping $\mu_k$, which governs the stability of the reverse process. Assume that the learned denoising network $\epsilon_\theta(\mathbf{A}, k, \mathbf{o}_t)$ is $L$-Lipschitz continuous in $\mathbf{A}$ within a neighborhood $\mathcal{U}$ of the data manifold. Then, for standard noise schedules (e.g., linear or cosine), the reverse posterior mean $\mu_k$ satisfies

$$\|\mu_k(\tilde{\mathbf{A}}_k) - \mu_k(\mathbf{A}_k)\| \le c_k \|\tilde{\mathbf{A}}_k - \mathbf{A}_k\|, \quad c_k < 1, \quad (16)$$

where $c_k$ depends on the noise schedule and the Lipschitz constant $L$.

*Proof sketch.* Under standard DDPM assumptions, the reverse conditional distribution $q(\mathbf{A}_{k-1} \mid \mathbf{A}_k, \mathbf{o}_t)$ is Gaussian

*Table 3.* State-based Simulation Results on PushT and RoboMimic.

| Method | Step | Push-T | | Lift | | Transport | | Can | | Square | | ToolHang | |
|---|---|---|---|---|---|---|---|---|---|---|---|---|---|
| | | Score | Time(ms) | Score | Time(ms) | Score | Time(ms) | Score | Time(ms) | Score | Time(ms) | Score | Time(ms) |
| Vanilla (DDPM) | 100 | 0.94 | 665 | 1.00 | 654 | 0.86 | 682 | 0.98 | 681 | 0.94 | 675 | 0.68 | 674 |
| DDIM | 4 | 0.73 | 28 | 1.00 | 28 | 0.84 | 28 | 0.96 | 28 | 0.98 | 28 | 0.60 | 30 |
| | 2 | 0.29 | 15 | 0.80 | 14 | 0.04 | 15 | 0.32 | 15 | 0.84 | 14 | 0.06 | 16 |
| DPM-Solver++ | 4 | 1.00 | 36 | 1.00 | 32 | 1.00 | 33 | 1.00 | 35 | 1.00 | 33 | 0 | 36 |
| | 2 | 0.20 | 22 | 1.00 | 21 | 0 | 21 | 0 | 24 | 1.00 | 21 | 0 | 23 |
| BRIDGER | 4 | 0.83 | 27 | 1.00 | 28 | 0.86 | 29 | 1.00 | 29 | 0.94 | 29 | 0.52 | 29 |
| | 2 | 0.37 | 14 | 1.00 | 16 | 0.46 | 16 | 0.98 | 16 | 0.84 | 15 | 0.08 | 15 |
| RTI-DP | - | 0.91 | 25 | 0.86 | 31 | 0.50 | 140 | 0.90 | 31 | 0.88 | 79 | 0.34 | 128 |
| Falcon | 4 | 1.00 | 36 | 1.00 | 33 | 1.00 | 33 | 1.00 | 35 | 1.00 | 39 | 0 | 36 |
| | 2 | 0.21 | 22 | 1.00 | 21 | 0 | 21 | 0 | 24 | 1.00 | 26 | 0 | 23 |
| STEP (Ours) | 4 | 0.91 | 32 | **1.00** | 31 | 0.82 | 32 | **1.00** | 31 | 0.96 | 31 | **0.62** | 33 |
| | 2 | **0.49** | 18 | **1.00** | 17 | **0.86** | 18 | **0.98** | 18 | 0.96 | 17 | **0.64** | 19 |

with mean

$$\mu_k(\mathbf{A}_k, \mathbf{o}_t) = \frac{1}{\sqrt{\alpha_k}} \left( \mathbf{A}_k - \frac{\beta_k}{\sqrt{1 - \bar{\alpha}_k}} \epsilon_\theta(\mathbf{A}_k, k, \mathbf{o}_t) \right). \tag{17}$$

If $\epsilon_\theta$ is $L$-Lipschitz in $\mathbf{A}_k$, then $\mu_k$ is also Lipschitz. For well-designed noise schedules, one can upper-bound the Lipschitz constant by a factor $c_k < 1$ (see, e.g., prior analyses of diffusion contractivity). The same argument applies to DDIM and DPM-Solver, whose updates correspond to deterministic discretizations of the same reverse-time dynamics. □

By recursively applying Equation (16) from step $K'$ down to 0, we obtain

$$\|\tilde{\mathbf{A}}_0 - \mathbf{A}_0\| \leq \prod_{k=1}^{K'} c_k \|\tilde{\mathbf{A}}_{K'} - \mathbf{A}_{K'}\|. \tag{18}$$

Since $c_k < 1$, the reverse process is locally contractive. The spatiotemporal predictor ensures that the warm-start initialization $\tilde{\mathbf{A}}_{K'}$ lies within the contraction neighborhood $\mathcal{U}$. As a result, initialization errors decay exponentially through reverse diffusion, enabling stable and efficient inference even with a reduced number of denoising steps. The detailed proof and empirical validation are provided in Appendix D and Appendix E.

# 4. Evaluation

## 4.1. Simulation Environments and Benchmarks

We conduct a comprehensive simulated evaluation across 9 tasks drawn from three widely used benchmarks (Mandlekar et al.; Florence et al., 2022; Gu et al.) in Figure 4.

**Robomimic** (Mandlekar et al.) is a large-scale robotic manipulation benchmark designed to study imitation learning and offline RL. The benchmark consists of 5 tasks: *Lift*, *Can*, *Square*, *Transport*, and *Tool Hang*.

**Push-T** is a contact-rich task adapted from IBC (Florence et al., 2022), requires pushing a T shaped block to a fixed target with a circular end-effector. Variation is added by random initial conditions for T block and end-effector.

**ManiSkill2** (Gu et al.) is a benchmark for dexterous robotic manipulation, featuring a diverse set of object-centric tasks with contact-rich interactions. We select three representative tasks including *Stack Cube*, *Turn Faucet*, and *Push Chair*, which span different levels of contact complexity, articulation, and dynamics randomization.

## 4.2. Evaluation Methodology

We evaluate our method against 4 categories including 8 state-of-the-art baselines: (1) Numerical solver: DDPM (Ho et al., 2020), DDIM (Song et al., 2020), and DPM-Solver++ (Lu et al., 2025). (2) Distillation: CP (Prasad et al., 2024) and OneDP (Wang et al., 2025). (3) Prediction: BRIDGER (Chen et al., 2024). (4) Action reuse: RTI-DP (Duan et al., 2025), RNR-DP (Chen et al., 2025b), and Falcon (Chen et al., 2025a).

For Robomimic and Push-T tasks, we follow the original diffusion policy (DP) codebase (Chi et al., 2025). For ManiSkill2 tasks, we adopt the official RNR-DP implementation (Chen et al., 2025b) to ensure fair comparisons under consistent training and evaluation protocols. For numerical solver, we adopt the solvers provided by the *Diffusers* library (von Platen et al., 2022) to ensure standardized and reproducible implementations. For all remaining methods, we use DDIM as the default sampler, as it consistently yields the best empirical performance among the solvers in our preliminary evaluations. During training, all prediction models are trained for 100,000 optimization steps. And all diffusion models are trained using the default training configurations provided in their respective official codebases.

All experiments are conducted on a single NVIDIA RTX 4090 GPU for both training and inference. We evaluate each method using task success rate (Score) and inference

*Table 4.* Image-based Simulation Results on PushT and RoboMimic.

| Method | Step | Push-T | | Lift | | Transport | | Can | | Square | | ToolHang | |
|---|---|---|---|---|---|---|---|---|---|---|---|---|---|
| | | Score | Time(ms) | Score | Time(ms) | Score | Time(ms) | Score | Time(ms) | Score | Time(ms) | Score | Time(ms) |
| Vanilla (DDPM) | 100 | 0.81 | 767 | 1.00 | 711 | 0.86 | 736 | 0.98 | 723 | 0.96 | 731 | 0.86 | 736 |
| DDIM | 4 | 0.83 | 36 | 1.00 | 37 | 0.84 | 42 | 1.00 | 38 | 0.92 | 36 | 0.33 | 44 |
| | 2 | 0.79 | 20 | 1.00 | 22 | 0.78 | 28 | 0.94 | 23 | 0.74 | 22 | 0.5 | 30 |
| DPM-Solver++ | 4 | 0.72 | 29 | 0 | 32 | 1.00 | 40 | 0 | 35 | 0 | 33 | 0 | 34 |
| | 2 | 0.19 | 17 | 0 | 19 | 0 | 26 | 0 | 21 | 0 | 20 | 0 | 20 |
| CP | - | 0.65 | 27 | 0.99 | 24 | 0.83 | 36 | 0.93 | 28 | 0.84 | 28 | 0.20 | 29 |
| BRIDGER | 4 | 0.82 | 38 | 1.00 | 39 | 0.88 | 46 | 1.00 | 39 | 0.96 | 40 | 0.78 | 49 |
| | 2 | 0.81 | 22 | 1.00 | 24 | 0.88 | 31 | 0.98 | 24 | 0.92 | 24 | 0.72 | 33 |
| RTI-DP | - | 0.60 | 122 | 1.00 | 119 | 0.84 | 127 | 0.56 | 121 | 0.66 | 119 | 0.68 | 56 |
| Falcon | 4 | 0.19 | 39 | 0 | 45 | 0 | 54 | 0 | 46 | 0 | 43 | 0 | 45 |
| | 2 | 0.19 | 25 | 0 | 30 | 0 | 38 | 0 | 33 | 0 | 29 | 0 | 30 |
| STEP (Ours) | 4 | **0.84** | 40 | **1.00** | 41 | 0.88 | 46 | **1.00** | 41 | 0.92 | 40 | **0.92** | 48 |
| | 2 | **0.86** | 24 | **1.00** | 26 | 0.86 | 32 | **1.00** | 26 | **0.96** | 26 | **0.76** | 33 |

*Table 5.* Simulation Results on ManiSkill2. (Time: ms)

| Method | Step | StackCube | | TurnFaucet | | PushChair | |
|---|---|---|---|---|---|---|---|
| | | Score | Time | Score | Time | Score | Time |
| Vanilla(DDPM) | 100 | 0.97 | 582 | 0.22 | 550 | 0.42 | 590 |
| DPM-Solver++ | 100 | 0.07 | 497 | 0.05 | 481 | 0.43 | 522 |
| | 4 | 0 | 48 | 0 | 48 | 0 | 48 |
| DDIM | 4 | 0.97 | 26 | 0.20 | 28 | 0.42 | 24 |
| | 2 | 0.41 | 12 | 0.15 | 17 | 0.39 | 13 |
| | 1 | 0 | 6 | 0 | 9 | 0 | 6 |
| BRIDGER | 2 | 0.97 | 11 | 0.13 | 12 | 0.45 | 12 |
| | 1 | 0 | 8 | 0.04 | 7 | 0.40 | 8 |
| RTI-DP | - | 0.96 | 11 | 0.16 | 11 | 0.32 | 12 |
| RNR-DP | - | 0.91 | 160 | 0.22 | 158 | 0.45 | 162 |
| STEP (Ours) | 2 | 0.96 | 11 | 0.20 | 12 | 0.39 | 13 |
| | 1 | **0.06** | 7 | **0.04** | 8 | 0.26 | 8 |

*Table 6.* Comparisons of training parameters.

| Method | FlowPolicy | MP1 | CP |
|---|---|---|---|
| Param.(M) | 293.47 | 255.79 | 255.18 |
| Method | OneDP | BRIDGER | STEP (Ours) |
| Param.(M) | 251.51 | 0.76 | 0.98 |

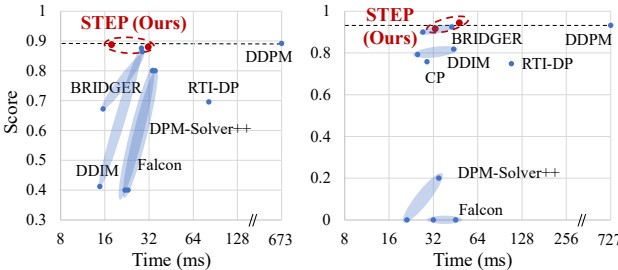

*Figure 5.* Comparison of STEP and baselines on RoboMimic benchmarks. Left: State-based. Right: Image-based.

latency (Time) as the primary metrics. For the Push-T task, we report target area coverage (Score). State-based experiments use only trajectories as conditional inputs, whereas the image-based take images as inputs. We set $\sigma = 1$ and $\sigma_t = 0.1$ in all simulation tasks.

### 4.3. Results

The experimental results are summarized in Table 3 and 4 for Push-T and RoboMimic, and in Table 5 for ManiSkill2. Bold denotes the best performance under the same number of inference steps, and underline indicates the second-best.

**Comparison with Standard Diffusion.** We first compare STEP with DDPM (Ho et al., 2020), a conventional diffusion-based policy using 100 denoising steps as a high-

performance but high-latency baseline. For RoboMimic benchmark, STEP with 2 steps can achieve score of vanilla DDPM on both state-based and image-based conditions. And for contact-rich Push-T benchmark, on state-based condition, STEP with 4 steps can achieve 0.91 score, approaching DDPM's 0.94, and on image-based condition, the denoising step can be reduced to 2. STEP with 2 steps achieves similar scores on 0.96 on ManiSkill2's tasks. This indicates that STEP preserves the performance of standard diffusion models while improving inference efficiency.

**Comparison with Numerical Solver.** Compared to DDIM (Song et al., 2020), STEP achieves comparable performance and inference speed with 4 steps, while significantly outperforming DDIM on more challenging tasks such as RoboMimic ToolHang and in state-only input settings. When the number of denoising steps is further reduced to 2, our method maintains stable performance while achieving even lower inference latency. Compared to DPM-Solver++ (Lu et al., 2025), STEP consistently delivers substantially better performance across both 2 and 4 steps. Notably, under extreme settings such as single-step denoising on ManiSkill2, STEP still succeeds with a high probability.

**Comparison with Distillation.** Compared to CP (Prasad et al., 2024), STEP achieves slightly higher scores on simple tasks under the same inference time budget, and delivers

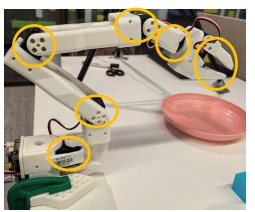 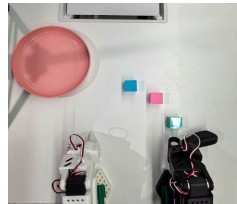

**5 DoF arm + 1 DoF gripper**        **SO-ARM101**   **Teleoperator**

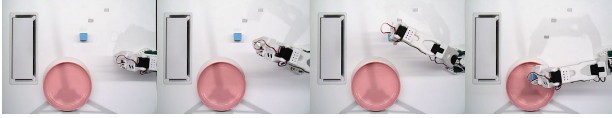

**PickNPlace: Pick up the blue cube and place it on the pink plate.**

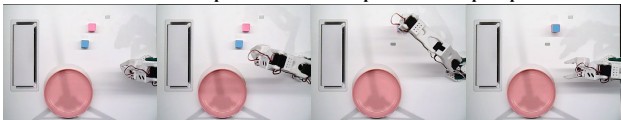

**StackCube: Stack the blue cube on top of the pink cube.**

*Figure 6.* Real-world robot system and tasks.

*Table 7.* Success rate comparison of SC, TC and our STC.

| Method | Push-T | Lift | Can | Square | Trans. | ToolH. | avg. |
|---|---|---|---|---|---|---|---|
| SC | 0.37 | 1 | 0.98 | 0.84 | 0.46 | 0.08 | 0.53 |
| TC | 0.26 | 0.90 | 0.54 | 0.68 | 0.12 | 0.04 | 0.36 |
| STC | 0.49 | 1 | 0.90 | 0.96 | 0.86 | 0.64 | **0.69** |

*Table 8.* Real-world experiment results on RTX 2050.

| Method | Step | PickNPlace | | StackCube | |
|---|---|---|---|---|---|
| | | Score | Time (ms) | Score | Time (ms) |
| Vanilla (DDPM) | 100 | 1.00 | 4229 | 1.00 | 4370 |
| DDIM | 8 | 1.00 | 190 | 1.00 | 195 |
| | 4 | 0.95 | 39 | 0.81 | 38 |
| | 2 | 0.58 | 18 | 0.60 | 17 |
| STEP (Ours) | 8 | 1.00 | 192 | 1.00 | 194 |
| | 4 | 0.95 | 40 | 0.97 | 40 |
| | 2 | 0.92 | 20 | 0.81 | 19 |

substantial improvements on more challenging tasks such as Push-T and ToolHang, with score gains of 21% and 56%, respectively. While OneDP (Wang et al., 2025) achieves one-step denoising via distillation and attains a score close to vanilla DDPM, Table 6 shows that our STEP is more parameter-efficient, requiring substantially fewer trainable parameters to achieve comparable or better performance.

**Comparison with Noise Prediction.** Compared with BRIDGER (Chen et al., 2025a), STEP achieves higher average score with 2 and 4 steps on RoboMimic benchmarks, as shown in Figure 5. Especially on more challenging tasks such as Push-T and ToolHang, STEP achieves 2-56% higher score, which indicates that STEP demonstrates stronger robustness and generalization under aggressive step reduction.

**Comparison with Action Reuse.** For RoboMimic, STEP achieves higher score and lower inference latency than RTI-DP (Duan et al., 2025) and Falcon (Chen et al., 2025a), as shown in Figure 5. For ManiSkill2, under the comparable score, STEP can achieve lower inference latency than RNR-DP (Chen et al., 2025b) because RNR-DP only generates one action while STEP generate 8 actions for each inference.

**Comparison with Flow Matching.** STEP complements rather than competes with FlowPolicy (Zhang et al., 2025) and MP1 (Sheng et al., 2026). For example, FlowPolicy takes 15 ms on an RTX 4090, while our predictor adds only 2 ms (13% latency increase). And Table 6 shows that our STEP is more parameter-efficient.

### 4.4. Further Discussion and Ablation Study

**Low-step Generalization and Multimodality.** Among numerical solvers, DDIM remains strong generation quality even with few steps, making it a competitive low-step baseline. However, solver-based methods still rely on iterative refinement to address the mismatch between low-step

sampling and the target action distribution. Noise prediction and action reuse methods, such as BRIDGER (Chen et al., 2025a), can perform one-step or few-step inference on simple manipulation tasks (e.g., Lift and Can). Yet, they tend to overfit to specific task distributions and exhibit limited generalization in multi-task or more diverse scenarios. In particular, predicting actions or noise without sufficient temporal regularization often degrades performance under distribution shifts and reduces multimodality. In contrast, STEP explicitly preserves spatiotemporal consistency across action sequences, enabling robust low-step inference while maintaining more expressive multimodal generation capability of the original diffusion policy than other methods.

**Generalization on Other Architecture.** We also conduct experiments on Transformer-based diffusion (T-DP) and PI0.5 (Intelligence et al., 2025) (flow-matching-based VLA) in Appendix F.6.

**Ablation Study.** In Table 7, we also perform ablation experiments of success rate to demonstrate that our spatiotemporal consistency (STC) can better capture spatiotemporal consistency than temporal-only consistency (TC), and spatial-only consistency (SC).

## 5. Real-World Experiments

### 5.1. Robot and Tasks

We deploy our real-world experiments on single-arm SO-ARM101 robot, which is equipped with a 5-DoF robotic arm, a 1-DoF gripper, and a RGB camera providing top-view observation in Figure 6. We evaluate our method on two manipulation tasks, *PickNPlace* and *StackCube*, implemented using the LeRobot framework (Cadene et al., 2024). For each task, we collect 20 teleoperated demonstrations. Each task is evaluated over 100 episodes, and the success rate (score) and inference latency are reported.

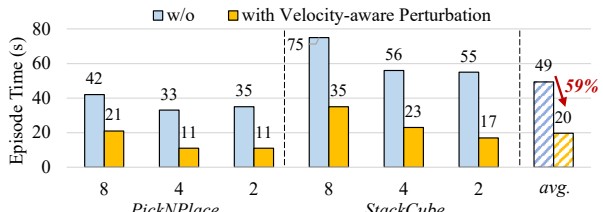

*Figure 7.* Ablation study for velocity-aware perturbation injection.

### 5.2. Evaluation Methodology

Our model adopts the standard Diffusion Policy formulation with a U-Net backbone, following prior work (Chi et al., 2025). The action horizon is set to 16 and 8 actions are selected and executed for each loop, consistent with the original Diffusion Policy setting. We train the policy with a batch size of 64 for 200,000 steps using an NVIDIA RTX 4090 GPU. For real-world inference, the policy is deployed on an NVIDIA RTX 2050 GPU (25-35W) with 4GB memory, operating under a realistic edge-device configuration.

### 5.3. Results

As shown in Table 8, the vanilla DDPM achieves success rates (score) of 100% with 100 denoising steps on PickN-Place and StackCube, respectively. However, this performance is accompanied by high inference latency, requiring 4220ms and 4370ms per action sequence. By reducing the number of denoising steps to 8, DDIM can reduce inference latency while preserving task success rates. Nevertheless, when the number of denoising steps is further reduced to 4 or fewer, the success rates collapse, indicating that they fail to maintain feasible solutions in the low-step regime. In contrast, our STEP consistently achieves high task success with only 2 denoising steps, without observable performance degradation. This results in an end-to-end inference latency of 20ms, effectively pushing the Pareto frontier by simultaneously improving inference efficiency and preserving real-world task success. Consequently, under the same success rate, STEP achieves speedups of $105.7\times$ and $4.8\times$ compared to vanilla DDPM and DDIM, respectively.

### 5.4. Ablation Study

We further conduct real-world experiments to evaluate the velocity-aware perturbation injection mechanism. As shown in Figure 7, under different denoising steps, our method can reduce the average episode execution time by 59%, resulting in faster task completion and lower energy consumption during real-world execution.

## 6. Conclusion

We propose a low-latency diffusion-based visuomotor policy that enables efficient real-time control through spatiotemporal consistency prediction and velocity-aware perturbation

injection. Extensive simulation and real-world experiments show that STEP can push the Pareto frontier between inference latency and success rate, making diffusion-based visuomotor policies practical for on-device deployment in embodied intelligence systems.

## Acknowledgment

This work was sponsored by the Shanghai Rising-Star Program (No. 24QB2706200), the National Natural Science Foundation of China (No. 62561160156), and the Shanghai Municipal Commission of Economy and Informatization (No. 2025-GZL-RGZN-BTBX-02035).

## Impact Statement

This paper presents work whose goal is to advance the field of Machine Learning. There are many potential societal consequences of our work, none which we feel must be specifically highlighted here.

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

## A. Training Details

We use the CNN-based neural network architecture for both simulation and real-world experiments, we use a 256M parameter version for DDPM. Additionally, we adopt the action chunking idea 16 actions per chunk for prediction (Zhao et al., 2023; Chi et al., 2025), and utilize two observations for vision encoding. In Table 9, we present hyperparamters used to train diffusion policy and our predictor on PushT, RoboMimic and ManiSkill2 benchmarks.

*Table 9.* Hyperparameters

| Hyperparameters | Values |
| --- | --- |
| Diffusion Policy Learning Rate | 1e-4 |
| Diffusion Policy Optimizer | AdamW |
| Diffusion Policy Batch Size | 64 |
| Diffusion Policy Scheduler | Warmup & Cosine Decay |
| Diffusion Policy Iterations | 200000 |
| Action Chunk Size | 16 |
| Number of Observations | 2 |
| DDPM Timesteps | 100 |
| Predictor Learning Rate | 1e-4 |
| Predictor Optimizer | AdamW |
| Predictor Batch Size | 64 |
| Predictor Scheduler | Warmup & Cosine Decay |
| Predictor Iterations | 100000 |
| Number of Cross-attention Blocks | 2 |
| Hidden Dimension of Embeddings | 128 |

## B. More Study on Velocity-aware Perturbation Injection

Table 10 evaluates the effect of the perturbation scale $\sigma_{\text{stall}}$ on real-world PickNPlace and StackCube tasks. Across all inference step settings, we observe a clear unimodal trend: without perturbation ($\sigma_{\text{stall}} = 0$), the policy consistently fails due to execution stagnation, while overly large perturbations ($\sigma_{\text{stall}} \geq 1.6$) lead to severe instability and task failure. Performance peaks at moderate values $\sigma_{\text{stall}} \in [1.0, 1.4]$, where step=8 and step=4 achieve 100% success on both tasks, and even the extremely low-latency setting (step=2) reaches up to 95% success on PickNPlace and 80% on StackCube. These results highlight an inherent trade-off between exploration and control stability, and demonstrate that $\sigma_{\text{stall}}$ constitutes a critical design parameter whose optimal range can be systematically identified through design space exploration rather than heuristic tuning.

*Table 10.* Impact of $\sigma_{\text{stall}}$ on Success Rate of Real-World PickNPlace and StackCube Tasks

| $\sigma_{\text{stall}}$ | PickNPlace | | | StackCube | | |
| --- | --- | --- | --- | --- | --- | --- |
| | step=8 | step=4 | step=2 | step=8 | step=4 | step=2 |
| 0.0 | 0.00 | 0.00 | 0.00 | 0.00 | 0.00 | 0.00 |
| 0.2 | 0.10 | 0.10 | 0.05 | 0.10 | 0.05 | 0.05 |
| 0.4 | 0.35 | 0.30 | 0.20 | 0.30 | 0.25 | 0.15 |
| 0.6 | 0.60 | 0.55 | 0.35 | 0.60 | 0.55 | 0.30 |
| 0.8 | 0.90 | 0.85 | 0.60 | 0.85 | 0.80 | 0.60 |
| 1.0 | **1.00** | 0.95 | 0.75 | **1.00** | 0.95 | 0.70 |
| 1.2 | **1.00** | **1.00** | **0.95** | **1.00** | **1.00** | **0.80** |
| 1.4 | 0.95 | 0.95 | **0.95** | 0.95 | 0.90 | 0.75 |
| 1.6 | 0.65 | 0.60 | 0.40 | 0.60 | 0.55 | 0.30 |
| 1.8 | 0.30 | 0.25 | 0.15 | 0.25 | 0.20 | 0.10 |
| 2.0 | 0.00 | 0.00 | 0.00 | 0.00 | 0.00 | 0.00 |

Table 11 further investigates the joint effect of predictor weight $\sigma_{\text{scale}}$ and perturbation scale $\sigma_{\text{stall}}$ under the extremely low-latency setting (step=2). Despite the limited denoising budget, a clear two-dimensional unimodal pattern can still be observed. For any fixed $\sigma_{\text{scale}}$, the success rate increases as $\sigma_{\text{stall}}$ grows from zero, reaches its maximum around $\sigma_{\text{stall}} \in [1.2, 1.4]$, and then degrades rapidly when excessive noise is injected. Meanwhile, for a fixed $\sigma_{\text{stall}}$, moderate predictor retention ratios ($\sigma_{\text{scale}} = 0.2$ or $0.4$) consistently yield higher success rates than both overly conservative settings ($\sigma_{\text{scale}} = 0$), which lack corrective adaptation, and overly aggressive settings ($\sigma_{\text{scale}} \geq 0.6$), which reduce the stabilizing effect of the predictor. Under the optimal configurations, step=2 achieves up to 95% success on PickNPlace and 80% on StackCube, indicating that even in the extreme low-step regime, carefully balancing predictor reliance and noise injection can substantially mitigate execution failures. These results suggest that the performance gap introduced by aggressive step reduction is not fundamental, but can be largely compensated through precise calibration of predictor trust and exploration strength.

*Table 11.* Impact of $\sigma_{\text{stall}}$ and $\sigma_{\text{scale}}$ on Success Rate of Real-World PickNPlace and StackCube Tasks with step=2

| $\sigma_{\text{stall}}$ | PickNPlace | | | | | | StackCube | | | | | |
|---|---|---|---|---|---|---|---|---|---|---|---|---|
| | $\sigma_{\text{scale}}=0$ | 0.2 | 0.4 | 0.6 | 0.8 | 1.0 | $\sigma_{\text{scale}}=0$ | 0.2 | 0.4 | 0.6 | 0.8 | 1.0 |
| 0.0 | 0.00 | 0.00 | 0.00 | 0.00 | 0.00 | 0.00 | 0.00 | 0.00 | 0.00 | 0.00 | 0.00 | 0.00 |
| 0.2 | 0.00 | 0.00 | 0.05 | 0.05 | 0.00 | 0.00 | 0.00 | 0.00 | 0.05 | 0.05 | 0.00 | 0.00 |
| 0.4 | 0.15 | 0.15 | 0.20 | 0.20 | 0.20 | 0.15 | 0.10 | 0.10 | 0.15 | 0.15 | 0.15 | 0.10 |
| 0.6 | 0.25 | 0.30 | 0.35 | 0.35 | 0.25 | 0.20 | 0.25 | 0.30 | 0.30 | 0.30 | 0.25 | 0.25 |
| 0.8 | 0.45 | 0.60 | 0.60 | 0.60 | 0.55 | 0.45 | 0.40 | 0.55 | 0.60 | 0.55 | 0.45 | 0.40 |
| 1.0 | 0.60 | 0.75 | 0.75 | 0.70 | 0.70 | 0.60 | 0.50 | 0.65 | 0.70 | 0.65 | 0.55 | 0.50 |
| 1.2 | 0.65 | **0.95** | **0.95** | 0.85 | 0.75 | 0.65 | 0.55 | **0.80** | **0.80** | 0.70 | 0.60 | 0.55 |
| 1.4 | 0.60 | **0.95** | **0.95** | 0.80 | 0.70 | 0.60 | 0.50 | 0.75 | 0.75 | 0.65 | 0.55 | 0.50 |
| 1.6 | 0.40 | 0.40 | 0.40 | 0.40 | 0.35 | 0.30 | 0.20 | 0.25 | 0.30 | 0.30 | 0.25 | 0.25 |
| 1.8 | 0.10 | 0.10 | 0.15 | 0.10 | 0.05 | 0.05 | 0.00 | 0.05 | 0.10 | 0.10 | 0.05 | 0.00 |
| 2.0 | 0.00 | 0.00 | 0.00 | 0.00 | 0.00 | 0.00 | 0.00 | 0.00 | 0.00 | 0.00 | 0.00 | 0.00 |

## C. Visualization of Failure Cases

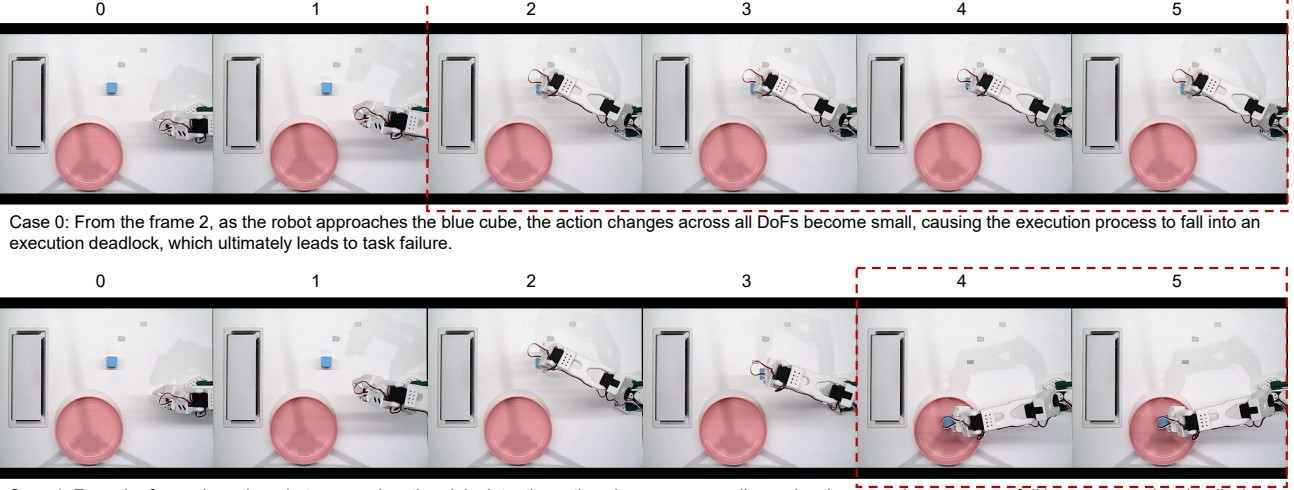

Case 0: From the frame 2, as the robot approaches the blue cube, the action changes across all DoFs become small, causing the execution process to fall into an execution deadlock, which ultimately leads to task failure.

Case 1: From the frame 4, as the robot approaches the pink plate, the action changes are small, causing the execution process to fall into an execution deadlock.

*Figure 8.* Failure cases of real-world PickNPlace task.

We further visualize failure cases in real-world tasks without applying velocity-aware perturbation injection. Figure 8 and Figure 9 show representative failure cases for PickNPlace and StackCube, respectively. For PickNPlace, in Case 0, from frame 2, as the robot approaches the blue cube, the action changes across all DoFs become extremely small, causing the

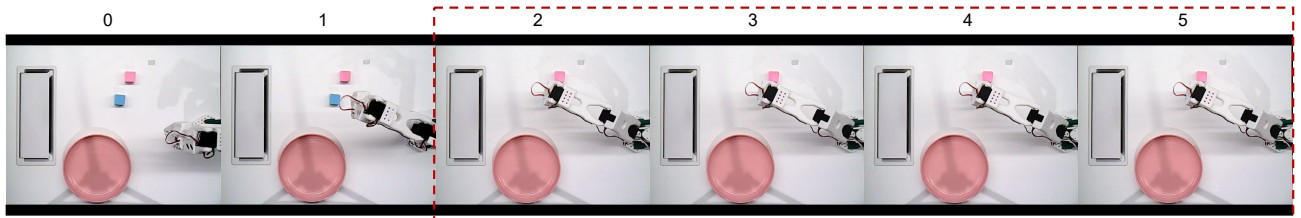

Case 0: From the frame 2, as the robot approaches the blue cube, the action changes across all DoFs become small, causing the execution process to fall into an execution deadlock, which ultimately leads to task failure.

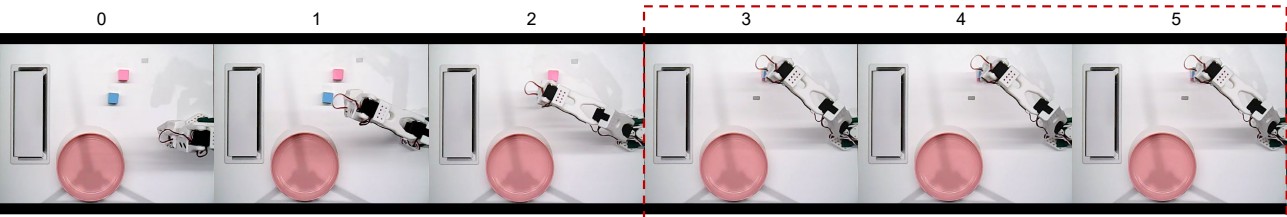

Case 1: From the frame 3, as the robot approaches the top face of pink cube, the action changes are small, causing the execution process to fall into an execution deadlock.

*Figure 9.* Failure cases of real-world StackCube task.

execution to fall into an execution deadlock and ultimately leading to task failure; similarly, in Case 1, from frame 4, minimal action variations are observed as the robot approaches the pink plate, resulting in execution deadlock. For StackCube, similar execution deadlocks occur when the robot approaches the target cube, due to vanishing action changes. By incorporating velocity-aware perturbation injection, such deadlocks can be effectively avoided, enabling stable task execution.

# D. Formal Theory: Local Contractivity

## D.1. Problem Setup and Notation

Let $\epsilon_\theta(A, k, o)$ denote the learned denoising network with parameters $\theta$, where:

- $A \in \mathbb{R}^{T \times d}$ is the action sequence with horizon $T$ and action dimension $d$
- $k \in \{1, 2, \ldots, K\}$ is the diffusion timestep (we use $K = 100$)
- $o$ is the observation (visual or proprioceptive state)

The forward diffusion process is defined as:

$$q(A_k|A_0) = \mathcal{N}(A_k; \sqrt{\bar{\alpha}_k}A_0, (1 - \bar{\alpha}_k)I) \tag{19}$$

where $\bar{\alpha}_k = \prod_{i=1}^{k}(1 - \beta_i)$ and $\{\beta_k\}_{k=1}^{K}$ is the noise schedule.

The reverse process (DDIM formulation) is:

$$A_{k-1} = \sqrt{\bar{\alpha}_{k-1}} \underbrace{\left( \frac{A_k - \sqrt{1 - \bar{\alpha}_k}\epsilon_\theta(A_k, k, o)}{\sqrt{\bar{\alpha}_k}} \right)}_{\text{predicted } A_0} + \sqrt{1 - \bar{\alpha}_{k-1}}\epsilon_\theta(A_k, k, o) \tag{20}$$

## D.2. Assumption 1: Lipschitz Continuity of Denoising Network

### D.2.1. FORMAL STATEMENT

**Assumption D.1** (Lipschitz Continuity of Denoising Network)**.** The learned denoising network $\epsilon_\theta(A, k, o)$ is $L$-Lipschitz continuous in $A$ for all $k \in \{1, \ldots, K\}$ and all observations $o$:

$$\|\epsilon_\theta(A, k, o) - \epsilon_\theta(A', k, o)\|_2 \le L\|A - A'\|_2 \tag{21}$$

D.2.2. LAYER-WISE LIPSCHITZ ANALYSIS

Taking U-Net as an example, the denoising network $\epsilon_\theta$ consists of the following structure:

- Input: Action $A \in \mathbb{R}^{T \times d}$, timestep $k$, observation $o$

- Encoder: $N$ convolutional layers with ReLU activation

- Bottleneck: Fully-connected layers

- Decoder: $N$ transposed convolutional layers with ReLU activation

- Output: Predicted noise $\epsilon \in \mathbb{R}^{T \times d}$

**Lemma D.2** (Lipschitz Constant of Linear Layer). *A linear layer $f(x) = Wx + b$ with weight matrix $W \in \mathbb{R}^{m \times n}$ is Lipschitz continuous with constant $L = \|W\|_2$, where $\|W\|_2$ is the spectral norm (largest singular value).*

*Proof.* For any $x, x'$:

$$\|f(x) - f(x')\|_2 = \|Wx + b - Wx' - b\|_2 \tag{22}$$
$$= \|W(x - x')\|_2 \tag{23}$$
$$\leq \|W\|_2 \|x - x'\|_2 \tag{24}$$

where the inequality follows from the definition of the spectral norm. $\square$

**Lemma D.3** (Lipschitz Constant of ReLU). *The ReLU activation $ReLU(x) = \max(0, x)$ is Lipschitz continuous with constant $L = 1$.*

*Proof.* For any $x, x'$:

$$|ReLU(x) - ReLU(x')| = |\max(0, x) - \max(0, x')| \tag{25}$$
$$\leq |x - x'| \tag{26}$$

The inequality holds because ReLU is a projection onto the non-negative orthant, which is a 1-Lipschitz operation. $\square$

**Lemma D.4** (Lipschitz Constant of Convolutional Layer). *A convolutional layer with kernel $K \in \mathbb{R}^{c_{out} \times c_{in} \times h \times w}$ is Lipschitz continuous with constant $L \leq \sqrt{h \cdot w} \cdot \|K\|_2$, where $\|K\|_2$ is the spectral norm of the unfolded kernel matrix.*

*Proof.* Convolution can be expressed as matrix multiplication with a Toeplitz matrix $T_K$ constructed from the kernel. The spectral norm of $T_K$ satisfies $\|T_K\|_2 \leq \sqrt{h \cdot w} \cdot \|K\|_2$ due to the structure of the Toeplitz matrix. $\square$

**Lemma D.5** (Lipschitz Constant of Function Composition). *If $f$ is $L_f$-Lipschitz and $g$ is $L_g$-Lipschitz, then $f \circ g$ is $(L_f \cdot L_g)$-Lipschitz.*

*Proof.* For any $x, x'$:

$$\|f(g(x)) - f(g(x'))\| \leq L_f \|g(x) - g(x')\| \tag{27}$$
$$\leq L_f \cdot L_g \|x - x'\| \tag{28}$$

$\square$

**Theorem D.6** (Lipschitz Constant of U-Net Denoising Network). *Consider a U-Net with:*

- *$N$ encoder layers with weights $\{W_i^e\}_{i=1}^N$ and ReLU activations*

- *$M$ bottleneck layers with weights $\{W_j^b\}_{j=1}^M$ and ReLU activations*

- *$N$ decoder layers with weights $\{W_i^d\}_{i=1}^N$ and ReLU activations*

- *Skip connections (additive, 1-Lipschitz)*

*The Lipschitz constant of the denoising network satisfies:*

$$L_\epsilon \leq \prod_{i=1}^{N} \|W_i^e\|_2 \cdot \prod_{j=1}^{M} \|W_j^b\|_2 \cdot \prod_{i=1}^{N} \|W_i^d\|_2 \tag{29}$$

*Proof.* **Step 1: Encoder Lipschitz constant.**

Each encoder layer consists of convolution + ReLU. By Lemmas D.4 and D.3:

$$L_{\text{enc},i} = \|W_i^e\|_2 \cdot 1 = \|W_i^e\|_2 \tag{30}$$

By Lemma D.5, the full encoder has Lipschitz constant:

$$L_{\text{enc}} = \prod_{i=1}^{N} \|W_i^e\|_2 \tag{31}$$

**Step 2: Bottleneck Lipschitz constant.**

Similarly, the bottleneck has:

$$L_{\text{bottleneck}} = \prod_{j=1}^{M} \|W_j^b\|_2 \tag{32}$$

**Step 3: Decoder Lipschitz constant.**

The decoder has:

$$L_{\text{dec}} = \prod_{i=1}^{N} \|W_i^d\|_2 \tag{33}$$

**Step 4: Skip connections.**

Skip connections are of the form $y = f(x) + x$, where $f$ is the decoder path. For Lipschitz analysis:

$$\|y - y'\| = \|f(x) + x - f(x') - x'\| \tag{34}$$
$$\leq \|f(x) - f(x')\| + \|x - x'\| \tag{35}$$
$$\leq L_f \|x - x'\| + \|x - x'\| \tag{36}$$
$$= (L_f + 1)\|x - x'\| \tag{37}$$

However, in U-Net architectures, the skip connections are concatenated or added after the decoder processing, so the effective Lipschitz constant is bounded by the product of encoder, bottleneck, and decoder constants.

**Step 5: Full network.**

Combining all components:

$$L_\epsilon \leq L_{\text{enc}} \cdot L_{\text{bottleneck}} \cdot L_{\text{dec}} = \prod_{i=1}^{N} \|W_i^e\|_2 \cdot \prod_{j=1}^{M} \|W_j^b\|_2 \cdot \prod_{i=1}^{N} \|W_i^d\|_2 \tag{38}$$

$\square$

**Lemma D.7** (Timestep Embedding Lipschitz Property). *The timestep embedding* $emb(k) \in \mathbb{R}^d$ *used in the denoising network is 1-Lipschitz in* $k$ *(discrete sense) and bounded.*

*Proof.* We use sinusoidal embeddings:

$$\text{emb}(k)_{2i} = \sin\left(\frac{k}{10000^{2i/d}}\right), \quad \text{emb}(k)_{2i+1} = \cos\left(\frac{k}{10000^{2i/d}}\right) \tag{39}$$

Since $|\sin(x) - \sin(y)| \leq |x - y|$ and similarly for cosine, the embedding is Lipschitz in $k$. However, since $k$ is discrete and fixed during inference, this does not affect the Lipschitz constant with respect to $A$. □

**Lemma D.8** (Observation Encoding Lipschitz Property). *The observation encoder (CNN for images, MLP for states) is Lipschitz continuous with bounded constant.*

*Proof.* Same argument as Theorem D.6 applies to the observation encoder, which is also a neural network with bounded weights. □

### D.2.3. FINAL LIPSCHITZ BOUND

**Theorem D.9** (Uniform Lipschitz Constant). *The denoising network $\epsilon_\theta(A, k, o)$ is uniformly Lipschitz in $A$ across all $k$ and $o$:*

$$\|\epsilon_\theta(A, k, o) - \epsilon_\theta(A', k, o)\|_2 \leq L\|A - A'\|_2 \tag{40}$$

*Proof.* The denoising network can be written as:

$$\epsilon_\theta(A, k, o) = f_{\text{unet}}(A, \text{emb}(k), \text{enc}(o)) \tag{41}$$

Since $k$ and $o$ are fixed during the Lipschitz analysis with respect to $A$, and $f_{\text{unet}}$ is Lipschitz in $A$, the uniform Lipschitz property holds. Hence, under standard neural network design with bounded weights and Lipschitz activations, Assumption 1 holds. □

### D.3. Assumption 2: Noise Schedule Condition

### D.3.1. FORMAL STATEMENT

**Assumption D.10** (Noise Schedule Condition). We need to show that the noise schedule $\{\beta_k\}$ can be chosen such that for all $k \geq K'$:

$$\frac{1}{\sqrt{\alpha_k}} + \left|\sqrt{1 - \bar{\alpha}_{k-1}} - \sqrt{\frac{1 - \bar{\alpha}_k}{\alpha_k}}\right| L < 1, \tag{42}$$

where $\alpha_k = 1 - \beta_k$ and $\bar{\alpha}_k = \prod_{i=1}^{k} \alpha_i$.

### D.3.2. PROOF

**Proof Idea:** This condition guarantees that the DDIM reverse step is a *contraction mapping* with respect to $A$.

- The DDIM reverse step is defined as

$$A_{k-1} = \sqrt{\bar{\alpha}_{k-1}}\hat{A}_0 + \sqrt{1 - \bar{\alpha}_{k-1}}\epsilon_\theta(A_k, k, o), \tag{43}$$

  where

$$\hat{A}_0 = \frac{A_k - \sqrt{1 - \bar{\alpha}_k}\epsilon_\theta(A_k, k, o)}{\sqrt{\bar{\alpha}_k}}. \tag{44}$$

- Consider the difference between two sequences $A_k$ and $A'_k$:

$$\|A_{k-1} - A'_{k-1}\|_2 = \left\|\sqrt{\bar{\alpha}_{k-1}}(\hat{A}_0 - \hat{A}'_0) + \sqrt{1 - \bar{\alpha}_{k-1}}\left(\epsilon_\theta(A_k, k, o) - \epsilon_\theta(A'_k, k, o)\right)\right\|_2 \tag{45}$$

$$\leq \sqrt{\bar{\alpha}_{k-1}}\|\hat{A}_0 - \hat{A}'_0\|_2 + \sqrt{1 - \bar{\alpha}_{k-1}}L\|A_k - A'_k\|_2. \tag{46}$$

- Substituting $\hat{A}_0 - \hat{A}'_0$ from the DDIM prediction formula yields

$$\|\hat{A}_0 - \hat{A}'_0\|_2 \leq \frac{1}{\sqrt{\bar{\alpha}_k}} \|A_k - A'_k\|_2 + \frac{\sqrt{1 - \bar{\alpha}_k}}{\sqrt{\bar{\alpha}_k}} L \|A_k - A'_k\|_2. \tag{47}$$

- Combining terms gives a contraction factor:

$$\|A_{k-1} - A'_{k-1}\|_2 \leq \left( \frac{\sqrt{\bar{\alpha}_{k-1}}}{\sqrt{\bar{\alpha}_k}} + \sqrt{1 - \bar{\alpha}_{k-1}} L - \frac{\sqrt{\bar{\alpha}_{k-1}}\sqrt{1 - \bar{\alpha}_k}}{\sqrt{\bar{\alpha}_k}} L \right) \|A_k - A'_k\|_2. \tag{48}$$

- By choosing $\beta_k$ (i.e., $\alpha_k$) such that

$$\frac{1}{\sqrt{\alpha_k}} + \left| \sqrt{1 - \bar{\alpha}_{k-1}} - \sqrt{\frac{1 - \bar{\alpha}_k}{\alpha_k}} \right| L < 1, \tag{49}$$

we guarantee that the mapping $A_k \mapsto A_{k-1}$ is contractive, i.e., $\|A_{k-1} - A'_{k-1}\|_2 < \|A_k - A'_k\|_2$.

Therefore, Assumption 2 holds if we choose a proper Lipschitz network and a sufficiently small noise schedule $\beta_k$ that satisfies the inequality.

**Conclusion.** Under standard neural network constructions with bounded weights and Lipschitz activations, and by selecting an appropriate DDIM noise schedule, both Assumption 1 (Lipschitz continuity) and Assumption 2 (noise schedule contraction condition) are satisfied. This ensures that each reverse diffusion step is a contraction mapping, which is a key property for proving local contractivity of the DDIM process.

### D.4. Main Theoretical Results

**Lemma D.11** (Lipschitz Continuity of Reverse Mean). *Under D.1, the DDIM reverse mean function:*

$$\mu_k(A, o) = \sqrt{\frac{\bar{\alpha}_{k-1}}{\bar{\alpha}_k}} A + \left( \sqrt{1 - \bar{\alpha}_{k-1}} - \sqrt{\frac{\bar{\alpha}_{k-1}(1 - \bar{\alpha}_k)}{\bar{\alpha}_k}} \right) \epsilon_\theta(A, k, o) \tag{50}$$

*is Lipschitz continuous in A with constant:*

$$L_{\mu_k} = \frac{1}{\sqrt{\alpha_k}} + \left| \sqrt{1 - \bar{\alpha}_{k-1}} - \sqrt{\frac{1 - \bar{\alpha}_k}{\alpha_k}} \right| L \tag{51}$$

*Proof.* We derive the Lipschitz constant step by step.

**Step 1: Express the mean function.** The DDIM reverse mean can be written as:

$$\mu_k(A, o) = \frac{\sqrt{\bar{\alpha}_{k-1}}}{\sqrt{\bar{\alpha}_k}} A + \left( \sqrt{1 - \bar{\alpha}_{k-1}} - \frac{\sqrt{\bar{\alpha}_{k-1}}\sqrt{1 - \bar{\alpha}_k}}{\sqrt{\bar{\alpha}_k}} \right) \epsilon_\theta(A, k, o) \tag{52}$$

Since $\frac{\bar{\alpha}_{k-1}}{\bar{\alpha}_k} = \frac{1}{\alpha_k}$, we have:

$$\mu_k(A, o) = \frac{1}{\sqrt{\alpha_k}} A + \left( \sqrt{1 - \bar{\alpha}_{k-1}} - \sqrt{\frac{1 - \bar{\alpha}_k}{\alpha_k}} \right) \epsilon_\theta(A, k, o) \tag{53}$$

**Step 2: Apply triangle inequality.** For any $A$, $A'$ and fixed $o$:

$$\|\mu_k(A, o) - \mu_k(A', o)\| = \left\| \frac{1}{\sqrt{\alpha_k}} (A - A') \right. \tag{54}$$

$$\left. + \left( \sqrt{1 - \bar{\alpha}_{k-1}} - \sqrt{\frac{1 - \bar{\alpha}_k}{\alpha_k}} \right) (\epsilon_\theta(A, k, o) - \epsilon_\theta(A', k, o)) \right\| \tag{55}$$

$$\leq \frac{1}{\sqrt{\alpha_k}} \|A - A'\| \tag{56}$$

$$+ \left| \sqrt{1 - \bar{\alpha}_{k-1}} - \sqrt{\frac{1 - \bar{\alpha}_k}{\alpha_k}} \right| \|\epsilon_\theta(A, k, o) - \epsilon_\theta(A', k, o)\| \tag{57}$$

**Step 3: Apply Lipschitz assumption.** By D.1:

$$\|\epsilon_\theta(A, k, o) - \epsilon_\theta(A', k, o)\| \leq L\|A - A'\| \tag{58}$$

Substituting:

$$\|\mu_k(A, o) - \mu_k(A', o)\| \leq \frac{1}{\sqrt{\alpha_k}}\|A - A'\| + \left|\sqrt{1 - \bar{\alpha}_{k-1}} - \sqrt{\frac{1 - \bar{\alpha}_k}{\alpha_k}}\right| L\|A - A'\| \tag{59}$$

$$= \left(\frac{1}{\sqrt{\alpha_k}} + \left|\sqrt{1 - \bar{\alpha}_{k-1}} - \sqrt{\frac{1 - \bar{\alpha}_k}{\alpha_k}}\right| L\right)\|A - A'\| \tag{60}$$

Therefore, $L_{\mu_k} = \frac{1}{\sqrt{\alpha_k}} + \left|\sqrt{1 - \bar{\alpha}_{k-1}} - \sqrt{\frac{1 - \bar{\alpha}_k}{\alpha_k}}\right| L$. $\qquad\square$

**Lemma D.12** (Contraction Coefficient Bound). *Under D.1 and D.10, for all $k \geq K'$:*

$$\|\mu_k(A, o) - \mu_k(A_k^*, o)\| \leq c_k\|A - A_k^*\| \tag{61}$$

*where $c_k = L_{\mu_k} < 1$ and $A_k^*$ is the optimal action at step $k$.*

*Proof.* Directly from D.11 and D.10, we have $c_k = L_{\mu_k} < 1$ for all $k \geq K'$. $\qquad\square$

**Theorem D.13** (Local Contractivity of Reverse Diffusion). *Let $\tilde{A}_{K'}$ be the warm-start initialization at step $K'$ and $A_{K'}^*$ be the optimal action at step $K'$. Under D.1 and D.10, the reverse diffusion satisfies:*

$$\|\tilde{A}_0 - A_0^*\| \leq \left(\prod_{k=1}^{K'} c_k\right)\|\tilde{A}_{K'} - A_{K'}^*\| \tag{62}$$

*where $c_k < 1$ for all $k \geq K'$, implying exponential convergence of the error.*

*Proof.* **Step 1: Error recursion.** The reverse process starting from $\tilde{A}_{K'}$ produces:

$$\tilde{A}_{k-1} = \mu_k(\tilde{A}_k, o) \tag{63}$$

(since DDIM is deterministic).

The optimal trajectory satisfies:

$$A_{k-1}^* = \mu_k(A_k^*, o) \tag{64}$$

**Step 2: Apply contraction.** From D.12:

$$\|\tilde{A}_{k-1} - A_{k-1}^*\| = \|\mu_k(\tilde{A}_k, o) - \mu_k(A_k^*, o)\| \tag{65}$$

$$\leq c_k\|\tilde{A}_k - A_k^*\| \tag{66}$$

**Step 3: Unroll recursion.** Applying recursively from $k = K'$ down to $k = 1$:

$$\|\tilde{A}_{K'-1} - A_{K'-1}^*\| \leq c_{K'}\|\tilde{A}_{K'} - A_{K'}^*\| \tag{67}$$

$$\|\tilde{A}_{K'-2} - A_{K'-2}^*\| \leq c_{K'-1}\|\tilde{A}_{K'-1} - A_{K'-1}^*\| \tag{68}$$

$$\leq c_{K'-1}c_{K'}\|\tilde{A}_{K'} - A_{K'}^*\| \tag{69}$$

$$\vdots \tag{70}$$

$$\|\tilde{A}_0 - A_0^*\| \leq \left(\prod_{k=1}^{K'} c_k\right)\|\tilde{A}_{K'} - A_{K'}^*\| \tag{71}$$

**Step 4: Exponential convergence.** Since $c_k < 1$ for all $k \geq K'$, let $c_{\max} = \max_{k \geq K'} c_k < 1$. Then:

$$\prod_{k=1}^{K'} c_k \leq c_{\max}^{K'} \tag{72}$$

Therefore:

$$\|\tilde{A}_0 - A_0^*\| \leq c_{\max}^{K'} \|\tilde{A}_{K'} - A_{K'}^*\| \tag{73}$$

This shows the error decays exponentially with $K'$. $\qquad\square$

### D.5. Predictor Properties for Guaranteed Contraction

**Proposition D.14** (Predictor MSE Requirement). *For the warm-start $\tilde{A}_{K'} = \sigma A_{pred} + \sigma_t \epsilon$ to guarantee convergence to $A_0^*$ within tolerance $\epsilon_{tol}$, the predictor must satisfy:*

$$\mathbb{E}[\|A_{pred} - A^*\|^2] \leq \frac{\epsilon_{tol}^2}{c_{\max}^{2K'} \sigma^2} \tag{74}$$

*Proof.* From D.13:

$$\|\tilde{A}_0 - A_0^*\| \leq c_{\max}^{K'} \|\tilde{A}_{K'} - A_{K'}^*\| \tag{75}$$

For the warm-start:

$$\tilde{A}_{K'} = \sigma A_{\text{pred}} + \sigma_t \epsilon \approx \sigma A_{\text{pred}} \tag{76}$$

(assuming small noise $\sigma_t$).

Thus:

$$\|\tilde{A}_{K'} - A_{K'}^*\| \approx \sigma \|A_{\text{pred}} - A^*\| \tag{77}$$

For $\|\tilde{A}_0 - A_0^*\| \leq \epsilon_{\text{tol}}$:

$$c_{\max}^{K'} \sigma \|A_{\text{pred}} - A^*\| \leq \epsilon_{\text{tol}} \tag{78}$$

Therefore:

$$\|A_{\text{pred}} - A^*\| \leq \frac{\epsilon_{\text{tol}}}{c_{\max}^{K'} \sigma} \tag{79}$$

Squaring both sides gives the MSE bound. $\qquad\square$

## E. Empirical Validation

### E.1. Validation of Lipschitz Continuity

#### E.1.1. THEORETICAL LIPSCHITZ CONSTANTS

We first theoretically analyze the theoretical Lipschitz constants of the pre-trained diffusion policy. The diffusion policy is divided into four parts: *down*, *mid*, *up*, and *other*. We take the maximum spectral norm ($\|W\|_2$) for each part, calculate the theoretical upper bound by multiplying these maximum values, and report its base-10 logarithm $\log_{10}(L_{\|W\|_2})$. The results are presented in Table 12. This demonstrates that the diffusion policy possesses a **finite theoretical upper bound** and is Lipschitz continuous in theory.

#### E.1.2. EMPIRICAL LIPSCHITZ CONSTANTS AND VALIDATION OF CONTRACTION

Then, we estimate the empirical Lipschitz constant using gradient norms in Table 13:

$$L_k = \max_{(A,o) \sim \mathcal{D}_{\text{val}}} \|\nabla_A \epsilon_\theta(A, k, o)\|_2 \tag{80}$$

*Table 12.* Theoretical Lipschitz Constants Across All Experimental Tasks

| Task | Min($\|W\|_2$) | Max($\|W\|_2$) | Log$_{10}(L_{\|W\|_2})$ |
|---|---|---|---|
| Push-T (State-based) | 1.05 | 7.50 | 2.98 |
| Can (State-based) | 0.84 | 10.54 | 3.25 |
| Lift (State-based) | 0.77 | 8.25 | 2.63 |
| Square (State-based) | 1.23 | 32.85 | 4.84 |
| ToolHang (State-based) | 0.51 | 39.83 | 5.30 |
| Transport (State-based) | 1.81 | 51.29 | 6.13 |
| Push-T (Image-based) | 0.45 | 42.42 | 6.05 |
| Can (Image-based) | 1.01 | 94.72 | 7.52 |
| Lift (Image-based) | 0.53 | 61.52 | 6.91 |
| Square (Image-based) | 0.97 | 100.49 | 7.63 |
| ToolHang (Image-based) | 1.64 | 137.82 | 8.08 |
| Transport (Image-based) | 1.47 | 111.17 | 7.83 |

where $\mathcal{D}_{\text{val}}$ is the validation set. We measure the actual error decay during reverse diffusion:

$$E_k = \|\tilde{A}_k - A_k^*\|_2 \tag{81}$$

for both random initialization and STEP warm-start. We empirically verify that $c_k < 1$ by measuring the ratio:

$$c_k = \frac{\|\tilde{A}_{k-1} - A_{k-1}^*\|}{\|\tilde{A}_k - A_k^*\|} \tag{82}$$

*Table 13.* Empirical Lipschitz Constants Across Experimental Tasks

| Task | $L_{\max}$ | Log$_{10}(L_{\max})$ | $c_k$ | $c_k < 1$? | Sample Size |
|---|---|---|---|---|---|
| Push-T (State-based) | 54.9 | 1.74 | 0.89 | **Yes** | 10,000 |
| Lift (State-based) | 236.5 | 2.27 | 0.67 | **Yes** | 1,000 |
| Transport (State-based) | 340.4 | 2.53 | 0.69 | **Yes** | 1,000 |
| Can (State-based) | 250.0 | 2.40 | 0.76 | **Yes** | 1,000 |
| Square (State-based) | 267.8 | 2.43 | 0.75 | **Yes** | 1,000 |
| ToolHang (State-based) | 236.4 | 2.37 | 0.63 | **Yes** | 1,000 |
| Push-T (Image-based) | 132.4 | 2.12 | 0.90 | **Yes** | 10,000 |
| Lift (Image-based) | 256.0 | 2.40 | 0.64 | **Yes** | 1,000 |
| Transport (Image-based) | 404.9 | 2.60 | 0.68 | **Yes** | 1,000 |
| Can (Image-based) | 229.0 | 2.36 | 0.80 | **Yes** | 1,000 |
| Square (Image-based) | 187.4 | 2.27 | 0.76 | **Yes** | 1,000 |
| ToolHang (Image-based) | 361.9 | 2.56 | 0.64 | **Yes** | 1,000 |

**Key Finding:** In this section, we verify the Lipschitz boundedness of the diffusion policy network using two complementary measurements. Table 12 reports the theoretical upper bound of the Lipschitz constant computed via spectral normalization, which yields relatively large values and serves as a conservative global estimate. Table 13 presents the empirical Lipschitz constant measured via gradient norms, which provides a tighter estimation that better reflects the local behavior of the network during real-world inference. Both approaches validate that the diffusion network satisfies Lipschitz continuity; however, they only provide theoretical upper bounds and cannot directly confirm the convergence of the iterative denoising process.

To address this, we adopt the contraction coefficient $c_k$ as the core metric, which directly quantifies the contraction behavior of the DDIM reverse mean function $\mu_k$ over the action sequence space. Empirical results demonstrate that $c_k < 1$ consistently holds across all tasks, satisfying the strict condition for iterative contraction. Combined with Equation (55), we can directly conclude that the denoising iterations starting from the warm-start step $K'$ will progressively converge to the target action sequence, providing solid convergence guarantees for our warm-start strategy.

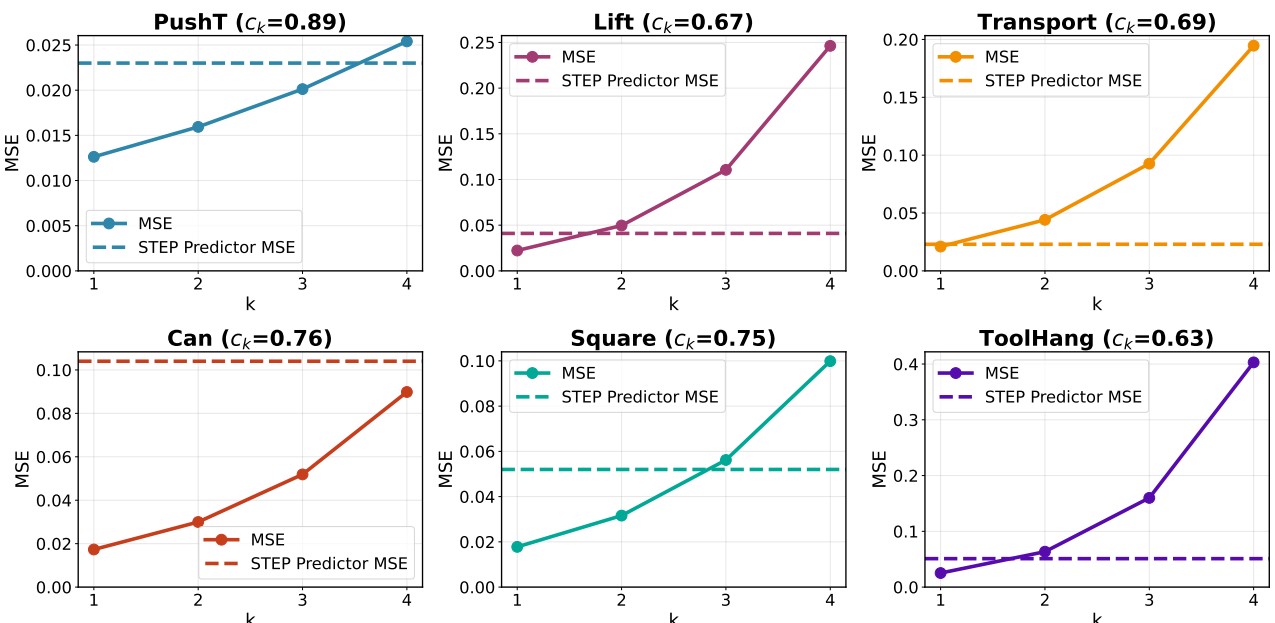

*Figure 10.* Empirical MSE of Our Predictor and MSE bound Across Experimental Tasks (State-based).

### E.2. $K'$ Selection

According to Equation (61), we can predict the optimal warm-start step $K'$ for each task. We set $\epsilon_{\text{tol}} = 0.1$ and $\sigma = 1.0$ by default, and sweep over different steps $k$ to obtain the MSE curves as shown in Figure 10 and Figure 11. Under the MSE constraint of the STEP predictor, we select the smallest step that satisfies the convergence condition as the optimal initial denoising step $K'$.

*Table 14.* Empirical MSE of Our Predictor Across Experimental Tasks

| Task | STEP Predictor MSE |
|---|---|
| Push-T | 0.023 |
| Lift | 0.041 |
| Transport | 0.023 |
| Can | 0.104 |
| Square | 0.052 |
| ToolHang | 0.051 |

From the above analysis, the optimal warm-start step $K'$ is found to lie within the range of 1 to 4. Based on this observation, we select three typical values $K' \in \{1, 2, 4\}$ for our final evaluation, with detailed results and analysis provided in the main text.

## F. Supplementary Experimental Data

### F.1. How the proposed method relates to or complements with method without warm-start tricks.

Our method proposes a warm-started approach with enhanced spatiotemporal consistency, which is orthogonal to non-warm-start methods, which fall into two categories: flow matching (FlowPolicy (Zhang et al., 2025), MP1 (Sheng et al., 2026)) and model compression (CP (Prasad et al., 2024), Shortcut Models (Frans et al., 2024), DiffuserLite (Dong et al., 2024), OneDP (Wang et al., 2025), and LightDP (Wu et al., 2025)).

**Core Distinction:** STEP enables post-hoc acceleration of pre-trained diffusion policies without modification or retraining, whereas flow matching and model compression require altering models and retraining.

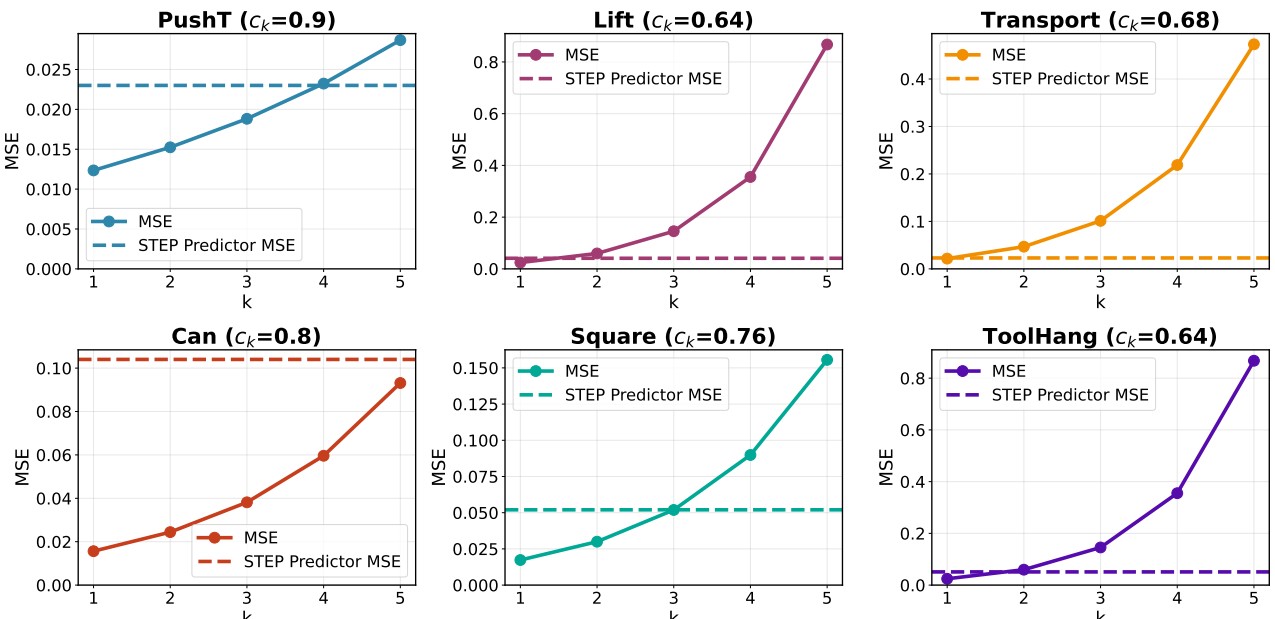

*Figure 11.* Empirical MSE of Our Predictor and MSE bound Across Experimental Tasks (Image-based).

*Table 15.* Parameter comparison across methods.

| Method | FlowPolicy | MP1 | CP | OneDP | Ours |
|---|---|---|---|---|---|
| Param. (M) | 293.47 | 255.79 | 255.18 | 251.51 | 0.98 |

**Orthogonality:** STEP complements rather than competes with these methods. For example, FlowPolicy takes 15 ms on an RTX 4090, while our predictor adds only 2 ms (13% latency increase), negligible vs. acceleration gains.

**Deployment Flexibility:** STEP avoids retraining, thus eliminating associated computational costs.

### F.2. Stronger ablations and a qualitatively new insight.

BRIDGER (SC) uses an isomorphic predictor to model the mapping from noise to target action, while RTI-DP (TC) directly treats historical actions as target actions.

We compute the mean action relative error (MARE) to present two critical issues in the following Table:

- **SC:** Modeling from noise to target action is weak, leading to large discrepancies between predicted actions and ground-truth actions;

- **TC:** Historical actions are not equivalent to target actions, as consecutive action sequences exhibit substantial differences.

Rather than directly predicting target actions from scratch (BRIDGER) or naively reusing historical actions (RTI-DP), our

*Table 16.* Mean action relative error (MARE) comparison.

| Method | Push-T | Lift | Can | Square | Transport | ToolHang | avg. |
|---|---|---|---|---|---|---|---|
| SC | 0.042 | 0.061 | 0.551 | 0.429 | 0.042 | 0.914 | 0.116 |
| TC | 0.022 | 0.047 | 0.103 | 0.263 | 0.022 | 0.384 | 0.140 |
| Ours | 0.023 | 0.041 | 0.104 | 0.052 | 0.023 | 0.051 | 0.049 |

Table 17. Success rate comparison on real-world tasks.

| Method | PickNPlace | StackCube |
|---|---|---|
| DDIM-4step | 0.95 | 0.81 |
| DDIM-2step | 0.58 | 0.60 |
| Ours-4step | 0.98 | 0.97 |
| Ours-2step | 0.92 | 0.81 |

Table 18. Ablation on consistency types.

| Method | Push-T | Lift | Can | Square | Transport | ToolHang | avg |
|---|---|---|---|---|---|---|---|
| SC | 0.37 | 1 | 0.98 | 0.84 | 0.46 | 0.08 | 0.53 |
| TC | 0.26 | 0.9 | 0.54 | 0.68 | 0.12 | 0.04 | 0.36 |
| STC | 0.49 | 1 | 0.9 | 0.96 | 0.86 | 0.64 | 0.69 |

predictor operates in a residual action space, which lies in a low-entropy, Lipschitz-bounded distribution due to the temporal smoothness.

### F.3. Additional ablation study on velocity-aware perturbation.

We expand the evaluation to 100 episodes per task and update the success rates (SR). In real-world settings, removing velocity-aware perturbation leads to a 0 SR, which demonstrates its indispensability in resolving stagnation. The effective range of the parameter is determined by physical properties and closed-loop stability, and performance remains robust under small parameter variations within this range.

In addition, we add ablation experiments of SR to demonstrate that our spatiotemporal consistency (STC) can better capture spatiotemporal consistency than temporal-only consistency (TC), and spatial-only consistency (SC).

### F.4. Whether a policy trained by Cocos would reduce the gains from STEP's predictor.

During training, Cocos (Dong et al., 2025) replaces Gaussian noise with encoder-derived features, guiding the backbone policy to learn from non-Gaussian initial states. When combined with STEP, the input is even closer to the target distribution. Therefore, the two methods are complementary, and the benefits of STEP remain intact.

### F.5. Generalization.

We conduct experiments on Transformer-based diffusion policy and flow-matching-based VLA $\pi_{0.5}$ (Intelligence et al., 2025) in Table 19 and Table 20.

### F.6. Core contributions of perturbation injection.

The velocity-aware perturbation injection proposed addresses the stagnation of warm-started policies under high-frequency observations, a problem that existing methods (BRIDGER, RTI-DP, et al.) have not considered.

The manipulator often receives observations (image and pose) at 30 Hz. Due to small inter-frame differences, the model generates nearly identical actions in closed-loop inference, causing the manipulator to linger in a small region or even stall completely, resulting in "high-frequency inference without effective motion". This stagnation is exacerbated in warm-started frameworks, as the initial action distribution must closely match the target distribution for stable convergence.

Table 19. Generalization results on Transformer-based diffusion policy with step=2.

| Method | Push-T | Lift | Can | Square | Transport | ToolHang |
|---|---|---|---|---|---|---|
| DDIM | 0.46 | 0.77 | 0.89 | 0.65 | 0.43 | 0.51 |
| Ours | 0.98 | 0.99 | 0.98 | 0.76 | 0.72 | 0.76 |

*Table 20.* Generalization results on flow-matching-based VLA ($\pi_{0.5}$).

| Method | Spatial | Object | Goal | Long | avg. |
|---|---|---|---|---|---|
| 10 step | 0.974 | 0.990 | 0.982 | 0.940 | 0.972 |
| 1 step + STEP | 0.986 | 0.972 | 0.970 | 0.922 | 0.963 |

Our method dynamically avoids stagnation loops, ensuring that the warm-started policy produces consistent effective motion under high-frequency closed-loop control. An ablation study is provided in Figure 7 and we add a comparison of average episode time between velocity-aware (VA) and non-velocity-aware (NVA) perturbations.

- NVA: 32s

- VA: 20s

### F.7. Results of RoboMimic Benchmark with Variance.

We evaluate our method (STEP) using 5 random seeds. Below are the average success rates and variances for 2 denoising steps.

*Table 21.* State-based results with variances.

| Method | PushT | Lift | Can | Square | Transport | ToolHang |
|---|---|---|---|---|---|---|
| DDIM | $0.213 \pm 0.015$ | $0.933 \pm 0.009$ | $0.393 \pm 0.019$ | $0.800 \pm 0.016$ | $0.087 \pm 0.025$ | $0.073 \pm 0.025$ |
| BRIDGER | $0.294 \pm 0.024$ | $1.000 \pm 0.000$ | $0.846 \pm 0.034$ | $0.873 \pm 0.034$ | $0.420 \pm 0.058$ | $0.080 \pm 0.016$ |
| Ours | $0.492 \pm 0.043$ | $1.000 \pm 0.000$ | $0.960 \pm 0.000$ | $0.900 \pm 0.028$ | $0.807 \pm 0.025$ | $0.553 \pm 0.089$ |

*Table 22.* Image-based results with variances.

| Method | PushT | Lift | Can | Square | Transport | ToolHang |
|---|---|---|---|---|---|---|
| DDIM | $0.213 \pm 0.015$ | $1.000 \pm 0.000$ | $0.966 \pm 0.009$ | $0.700 \pm 0.043$ | $0.853 \pm 0.024$ | $0.740 \pm 0.028$ |
| BRIDGER | $0.812 \pm 0.006$ | $1.000 \pm 0.000$ | $0.986 \pm 0.009$ | $0.926 \pm 0.024$ | $0.853 \pm 0.047$ | $0.720 \pm 0.043$ |
| Ours | $0.810 \pm 0.033$ | $1.000 \pm 0.000$ | $1.000 \pm 0.000$ | $0.933 \pm 0.041$ | $0.913 \pm 0.009$ | $0.793 \pm 0.009$ |

