# OpenReview forum: "STEP: Warm-Started Visuomotor Policies with Spatiotemporal Consistency Prediction"
_ICML.cc/2026/Conference — ICML 2026 regular_

### Official Review · Reviewer_2FB2 · 2026-03-11

**Soundness:** 3
**Presentation:** 2
**Significance:** 3
**Originality:** 2
**Overall Recommendation:** 4
**Confidence:** 3

**Summary:**

This paper aims to reduce the inference latency of diffusion-based visuomotor policies without sacrificing action quality. The authors propose STEP, which combines a transformer-based predictor that capture warm-starts diffusion from a predicted action sequence conditioned on the current observation and cached past actions, and a velocity-aware perturbation mechanism that injects actuation noise when stagnation is detected. The proposed method is evaluated on multiple simulated tasks and two real-world tasks, with reported gains in task performance and reduced latency over several accelerated diffusion baselines in few-step settings.

**Compliance With Llm Reviewing Policy:**

Affirmed.

**Final Justification:**

The authors addressed the majority of the concerns raised in the initial review. The additional results and the authors' clarifications have strengthened the submission. The strengths of this work now outweigh its weaknesses and limitations. All factors considered, I raise my score.

**Key Questions For Authors:**

See Strengths and Weaknesses.

**Limitations:**

Limitations such as potential issues of mode averaging were not discussed.

**Strengths And Weaknesses:**

## Strengths

1. It is well-motivated and important to study efficient diffusion-based policies with strong spatialtemporal consistency.
2. The proposed predictor is lightweight with minimal parameter overhead and appears to be computationally efficient.
3. The experimental evaluations are comprehensive, include reasonable baselines with both state- and image-based setting, and provide multiple ablation studies.
4. Real-world demonstrations with only edge devices are provided, demonstrating the practicality of the proposed method.


## Weaknesses

1. The contributions are limited. The perturbation injection mechanism does not appear to constitute a major contribution. The authors neither provide general insights or theoretical justification for this mechanism, nor ablation studies demonstrating it is critical for preserving spatiotemporal consistency or task performance. As such, it seems more like an engineering technique rather than an algorithmic innovation.
2. The MSE training objective (Eq. (12)) may be prone to mode averaging in multimodal settings and this issue is not discussed.
3. The local contractivity analysis in Sec. 3.4 is informal. There is no formal theorem or proposition statement, and the authors provide only a proof sketch without a complete formal proof in the appendix.
4. It is unclear what research questions the experimental sections aim to address, or what claims they are intended to support. Relying solely on task success rate as the evaluation metric is insufficient to empirically demonstrate that the proposed method better captures spatiotemporal consistency.
5. Variance is not reported for any of the quantitative results.
6. The paper does not discuss the limitations of the proposed approach.
7. Reproducibility: neither a code implementation nor an anonymous repository is provided.

---

> ### Author Rebuttal · Authors · 2026-03-26
>
> ### Q1: The contributions of perturbation injection are limited.
>
> **We respectfully disagree. The velocity-aware perturbation injection proposed addresses the stagnation of warm-started policies under high-frequency observations, a problem that existing methods (BRIDGER, RTI-DP, et. al) have not considered.**
>
> The manipulator often receives observations (image and pose) at 30 Hz. Due to small inter-frame differences, the model generates nearly identical actions in closed-loop inference, causing the manipulator to linger in a small region or even stall completely, resulting in **"high-frequency inference without effective motion"**. This stagnation is exacerbated in warm-started frameworks, as the initial action distribution must closely match the target distribution for stable convergence.
>
> Our method **dynamically avoid stagnation loops**, ensuring that the warm-started policy to produce consistent effective motion under high-frequency closed-loop control. An ablation study is provided in Figure 7 and we add a comparison of average episode time between velocity-aware (VA) and Non-Velocity-Aware (NVA) perturbations.
> - NVA: 32s
> - VA: 20s
>
> ### Q2: The MSE training objective may be prone to mode averaging in multimodal settings and this issue is not discussed.
>
> Thank you for raising this point. The MSE training strategy **follows the same strategy as the original Diffusion Policy**, which computes MSE loss between the predicted actions and the ground-truth actions.
> Both the predictor and original Diffusion Policy are models conditioned on observation $o_t$.  Therefore, different conditions lead to distinct and meaningful action distributions, thus avoiding undesired mode averaging.
>
> ### Q3: The local contractivity analysis in Sec. 3.4 is informal. There is no formal theorem or proposition statement, and the authors provide only a proof sketch without a complete formal proof in the appendix.
>
> We thank the reviewer for the helpful comment. We have added the **theoretical proof and experimental validation** of local contraction, which are provided in the anonymous link. https://anonymous.4open.science/r/STEP-1140/rebuttal/proof.pdf
>
> ### Q4: Only success rate is insufficient to demonstrate that the proposed method better captures spatiotemporal consistency.
>
> **We respectfully disagree. The experimental sections aim to validate that our method achieves a better Pareto frontier between success rate (SR) and inference time**, which is shown in Tables 2, 3, 4, 5 and Figure 5. In addition, we add **ablation experiments of SR, inference time and mean action relative error (MARE)** to demonstrate that spatiotemporal consistency (STC) can better capture spatiotemporal consistency than temporal-only consistency (TC), and spatial-only consistency (SC).
> - SR and time(ms)
> |Method|Push-T|Lift|Can|Square|Transport|ToolHang|avg. SR|avg. Time|
> |-|-|-|-|-|-|-|-|-|
> |SC|0.37|1|0.98|0.84|0.46|0.08|0.53|13|
> |TC|0.26|0.9|0.54|0.68|0.12|0.04|0.36|14|
> |STC(Ours)|0.49|1|0.9|0.96|0.86|0.64|0.69|15|
>
> - MARE
> |Method|Push-T|Lift|Can|Square|Transport|ToolHang|avg.|
> |-|-|-|-|-|-|-|-|
> |SC|0.419|0.361|0.551|4.029|0.419|12.914|3.116|
> |TC|0.022|0.047|0.103|0.263|0.022|0.384|0.140|
> |STC(Ours)|0.023|0.041|0.104|0.052|0.023|0.051|0.049|
>
>
> ### Q5: Variance is not reported.
>
> Thank you for the suggestion. We evaluate our method (STEP) using **5 random seeds**. Below are the average success rates and variances for 2 denoising steps.
> - State-based:
> |Method|PushT|Lift|Can|Square|Transport|ToolHang|
> |-|-|-|-|-|-|-|
> |DDIM|0.213±0.015|0.933±0.009|0.393±0.019|0.800±0.016|0.087±0.025|0.073±0.025|
> |BRIDGER|0.294±0.024|1.000±0.000|0.846±0.034|0.873±0.034|0.420±0.058|0.080±0.016|
> |Ours|0.492±0.043|1.000±0.000|0.960±0.000|0.900±0.028|0.807±0.025|0.553±0.089|
>
> - Image-based:
> |Method|PushT|Lift|Can|Square|Transport|ToolHang|
> |-|-|-|-|-|-|-|
> |DDIM|0.213±0.015|1.000±0.000|0.966±0.009|0.700±0.043|0.853±0.024|0.740±0.028|
> |BRIDGER|0.812±0.006|1.000±0.000|0.986±0.009|0.926±0.024|0.853±0.047|0.720±0.043|
> |Ours|0.810±0.033|1.000±0.000|1.000±0.000|0.933±0.041|0.913±0.009|0.793±0.009|
>
>
>
> ### Q6: Not discuss the limitations.
>
> Our approach relies on velocity-aware perturbation and is somewhat sensitive to its hyperparameters. In addition, the predictor requires task-specific training on a given dataset, similar to backbone policies such as diffusion policies and flow matching models. These limitations should be addressed in future work.
>
> ### Q7: Reproducibility.
>
> The code is open-sourced at: https://anonymous.4open.science/r/STEP-1140

---

> > ### Author Rebuttal · Reviewer_2FB2 · 2026-04-01
> >
> > The authors have addressed the majority of the raised concerns. The rebuttal substantially improves the quality of the submission. The authors should include the new results and contents with more details and analyses to the revised paper. Given this, I will adjust my rating.

---

> > > ### Author Response · Authors · 2026-04-03
> > >
> > > We are very glad to know that your concerns have been resolved! We sincerely thank you for your careful reading and constructive suggestions. We will supplement the clarified explanations and details in the revised paper. Thank you again for your time and effort in reviewing our work.

---

### Official Review · Reviewer_1AHr · 2026-03-12

**Soundness:** 2
**Presentation:** 2
**Significance:** 2
**Originality:** 3
**Overall Recommendation:** 4
**Confidence:** 4

**Summary:**

The paper proposes STEP, a lightweight warm-start mechanism for DDPM-based diffusion policies that initializes the reverse denoising process at an intermediate noise level using a cross-attention predictor conditioned on both current observations and historical actions. Evaluated across nine simulated benchmarks and two real-world tasks, the method reduces inference to 2 denoising steps while maintaining success rates competitive with 100-step DDPM baselines, with the main empirical story being a consistent improvement over BRIDGER and DDIM in the low-step regime.

**Compliance With Llm Reviewing Policy:**

Affirmed.

**Final Justification:**

I have decided to raise my score to 4 as the authors have substantially addressed the critical empirical gaps . The addition of formal convergence proofs and the empirical verification of the contractive constant also resolve my previous concerns regarding the theoretical grounding of the warm-start approach. However, I maintain that the technical novelty remains somewhat incremental, as the architecture essentially integrates spatiotemporal consistency concepts seen in prior work rather than proposing a fundamental paradigm shift in generative modeling. Furthermore, the velocity-aware perturbation mechanism exhibits high sensitivity to the σ hyperparameter, which raises questions about its robustness across diverse robotic platforms and environments. While the method provides a practical acceleration for DDPM-based policies, its long-term significance is somewhat limited by the field's rapid shift toward native one-step generation methods like Flow Matching.

**Key Questions For Authors:**

See weakness.

**Limitations:**

The authors' limitations section does not engage with the method's dependence on DDPM as the underlying policy class. The approach is only tested on a small U-Net backbone, and whether spatiotemporal warm-starting generalizes to Transformer-based diffusion policies, flow matching heads, or larger VLA-scale architectures is entirely open.

**Strengths And Weaknesses:**

Pros:

- The spatiotemporal consistency taxonomy in Section 3.1 provides a clean and useful framework for categorizing prior acceleration methods, and the two failure case analyses (temporal-only and spatial-only warm-starts) are well-motivated.

- The velocity-aware perturbation injection mechanism addresses a genuinely practical problem — execution deadlock under low-step denoising — that is often ignored in simulation-only work, and the real-world deployment on an edge GPU adds credibility to the latency claims.

- The simulation coverage across nine tasks from three benchmarks is broad, and the parameter efficiency comparison in Table 5 is a useful addition.

Cons:
- The baseline comparison is restricted to DDPM-era acceleration methods — DDIM, DPM-Solver++, BRIDGER, Falcon — which no longer reflects the competitive landscape at the time of submission. The broader community has produced a substantial body of work achieving far higher control frequencies without warm-start tricks, spanning architecture compression, flow matching, and consistency-based single-step generation (LightDP, FlowPolicy, Shortcut Models, DiffuserLite, MP1, OneDP). Against this backdrop, the paper's Pareto frontier claim is difficult to evaluate without situating the contribution within this wider context, and a discussion of how the proposed approach relates to or complements these directions would significantly strengthen the paper's positioning.
- The core technical contribution — jointly conditioning a lightweight predictor on current observations and historical actions to warm-start the reverse diffusion at an intermediate noise level — amounts to a direct and fairly natural combination of what BRIDGER and RTI-DP already do independently. BRIDGER conditions on observations for spatial alignment; RTI-DP reuses historical actions for temporal continuity. Doing both simultaneously is an incremental design decision, and the spatiotemporal consistency taxonomy in Section 3.1, while a clean organizing framework, is a post-hoc conceptual lens rather than a technical contribution in its own right. The paper would need substantially stronger ablations or a qualitatively new insight to argue that this combination produces behavior beyond the sum of its parts.
- The real-world evaluation is thin: two simple tabletop tasks, 20 episodes each, no statistical significance reporting. More critically, without the velocity-aware perturbation mechanism the real-world success rate is uniformly zero, yet σ_stall must be tuned within a narrow window ([1.2, 1.4]) to achieve peak performance — a sensitivity that raises serious questions about deployment generalizability. There is also no ablation isolating spatial-only versus temporal-only versus spatiotemporal conditioning, which is the most direct validation of the paper's central claim.
- The contractivity proof assumes L-Lipschitz continuity of the denoising network and c_k < 1 without empirical verification of either, deferring to "prior analyses" rather than providing a self-contained derivation. The theory neither predicts an optimal K' nor specifies what predictor properties are needed to guarantee the warm-start lands within the contraction neighborhood, making it unfalsifiable as stated.
- The proposed predictor shares a conceptual similarity with "Conditioning Matters" (Cocos, NeurIPS 2025) that the paper does not discuss — both reduce the transport distance between the generative starting point and the target action manifold using condition information, differing primarily in whether this is done at training time or inference time. It would be informative if the authors discussed how they understand this relationship, and in particular whether a policy trained with Cocos would reduce or eliminate the gains from STEP's predictor.

---

> ### Author Rebuttal · Authors · 2026-03-26
>
> ### Q1: Discuss how the proposed method relates to or complements with method without warm-start tricks.
>
> Thank you for raising this valuable point. Our method proposes a warm-started approach with enhanced spatiotemporal consistency, which is orthogonal to non-warm-start methods, which fall into two categories: flow matching (FlowPolicy/MP1) and model compression (CP/Shortcut Models/DiffuserLite/OneDP/LightDP).
>
> - **Core Distinction**: STEP enables **post-hoc acceleration** of pre-trained diffusion policies without modification or retraining, whereas flow matching and model compression require altering models and retraining.
>
> |Method|FlowPolicy|MP1|CP|OneDP|Ours|
> |-|-|-|-|-|-|
> |Param. (M)|293.47|255.79|255.18|251.51|0.98|
>
> - **Orthogonality**: STEP complements rather than competes with these methods. For example, FlowPolicy takes 15 ms on an RTX 4090, while our predictor adds only 2 ms (13% latency increase), negligible vs. acceleration gains.
>
> - **Deployment Flexibility**: STEP avoids retraining, thus eliminating associated computational costs.
>
> - **Theoretical Guarantees**: Theoretical and empirical justifications are provided in our rebuttal link in Q4.
>
>
> ### Q2: The paper would need stronger ablations or a qualitatively new insight to argue that this combination produces behavior beyond the sum of its parts.
>
> We would like to clarify that our method is **not the combination**.
> BRIDGER (SC) uses an isomorphic predictor to model the mapping from noise to target action, while RTI-DP (TC) directly treats historical actions as target actions.
>
> We compute the **mean action relative error (MARE)** to present two critical issues in the following Table:
> - SC: Modeling from noise to target action is weak, leading to large discrepancies between predicted actions and ground-truth actions;
> - TC: Historical actions are not equivalent to target actions, as consecutive action sequences exhibit substantial differences.
>
> |Method|Push-T|Lift|Can|Square|Transport|ToolHang|avg.|
> |-|-|-|-|-|-|-|-|
> |SC|0.042|0.061|0.551|0.429|0.042|0.914|0.116|
> |TC|0.022|0.047|0.103|0.263|0.022|0.384|0.140|
> |Ours|0.023|0.041|0.104|0.052|0.023|0.051|0.049|
>
> Rather than directly predicting target actions from scratch (BRIDGER) or naively reusing historical actions (RTI-DP), our predictor **operates in a residual action space, which lies in a low-entropy, Lipschitz-bounded distribution due to the temporal smoothness**.
>
>
> ### Q3: Concerns on velocity-aware perturbation and additional ablation study.
>
> We expand the evaluation to 100 episodes per task and update the success rates (SR). In real-world settings, removing velocity-aware perturbation leads to a 0 SR, which demonstrates its indispensability in resolving stagnation. The effective range of $σ_{stall}$ is determined by physical properties and closed-loop stability, and performance remains robust under small parameter variations within this range.
> |Method|PickNPlace|StackCube|
> |-|-|-|
> |DDIM-4step|0.95|0.81|
> |DDIM-2step|0.58|0.60|
> |Ours-4step|0.98|0.97|
> |Ours-2step|0.92|0.81|
>
> In addition, we add **ablation experiments of SR** to demonstrate that our spatiotemporal consistency (STC) can better capture spatiotemporal consistency than temporal-only consistency (TC), and spatial-only consistency (SC).
> |Method|Push-T|Lift|Can|Square|Transport|ToolHang|avg|
> |-|-|-|-|-|-|-|-|
> |SC|0.37|1|0.98|0.84|0.46|0.08|0.53|
> |TC|0.26|0.9|0.54|0.68|0.12|0.04|0.36|
> |STC|0.49|1|0.9|0.96|0.86|0.64|0.69|
>
> ### Q4: The assumption lacks empirical verification and discussion on optimal K.
>
> We provide explicit theoretical and empirical justifications in https://anonymous.4open.science/r/STEP-1140/rebuttal/proof.pdf:
>
> - Lipschitz continuity: We provide a self-contained theoretical proof that the diffusion policy is Lipschitz bounded, and empirically verify the denoising network’s Lipschitz continuity by estimating the explicit constant $L$ via gradient norms.
> - Contraction condition: We empirically confirm $c_k<1$.
> - Optimal K': We predict the optimal warm-start step K' is 1 to 4, which aligns with experimental results.
>
> ### Q5: Discuss whether a policy trained by Cocos would reduce the gains from STEP's predictor.
>
> Thanks for pointing out Cocos. During training, Cocos replaces Gaussian noise with encoder-derived features, guiding the backbone policy to learn from non-Gaussian initial states. When combined with STEP, the input is **even closer to the target distribution**. **Therefore, the two methods are complementary, and the benefits of STEP remain intact.**
>
> ### Q6: Generalization.
> Thanks for this valuable comment. We conduct experiments on Transformer-based diffusion (T-DP) and PI0.5 (flow-matching-based VLA).
>
> - T-DP (step=2)
> |Method|Push-T|Lift|Can|Square|Transport|ToolHang|
> |-|-|-|-|-|-|-|
> |DDIM|0.46|0.77|0.89|0.65|0.43|0.51|
> |Ours|0.98|0.99|0.98|0.76|0.72|0.76|
>
> - PI0.5
> |Method|Spatial|Object|Goal|Long|avg.|
> |-|-|-|-|-|-|
> |10step|0.974|0.990|0.982|0.940|0.972|
> |1step+Ours|0.986|0.972|0.970|0.922|0.963|

---

> > ### Author Rebuttal · Reviewer_1AHr · 2026-04-03
> >
> > I have decided to raise my score to 4 as the authors have substantially addressed the critical empirical gaps . The addition of formal convergence proofs and the empirical verification of the contractive constant also resolve my previous concerns regarding the theoretical grounding of the warm-start approach. However, I maintain that the technical novelty remains somewhat incremental, as the architecture essentially integrates spatiotemporal consistency concepts seen in prior work rather than proposing a fundamental paradigm shift in generative modeling. Furthermore, the velocity-aware perturbation mechanism exhibits high sensitivity to the σ hyperparameter, which raises questions about its robustness across diverse robotic platforms and environments. While the method provides a practical acceleration for DDPM-based policies, its long-term significance is somewhat limited by the field's rapid shift toward native one-step generation methods like Flow Matching. Nevertheless, given the improved empirical rigor and the method's clear utility, the work now meets the standard for acceptance.

---

> > > ### Author Response · Authors · 2026-04-05
> > >
> > > We are delighted that our revisions have successfully addressed your prior concerns, and we sincerely appreciate your meticulous review and insightful constructive comments.
> > > Your professional suggestions have significantly guided us in enhancing the empirical rigor and theoretical foundation of our work. We will meticulously supplement the corresponding clarifications and detailed supplementary contents in the revised manuscript to further perfect this paper.
> > > Once again, we express our deepest gratitude for your valuable time and dedicated effort in evaluating our research.

---

### Official Review · Reviewer_4LGY · 2026-03-13

**Soundness:** 3
**Presentation:** 3
**Significance:** 3
**Originality:** 3
**Overall Recommendation:** 4
**Confidence:** 3

**Summary:**

The paper introduces STEP, a framework designed to mitigate the high inference latency of diffusion-based policies in robotic manipulation. While diffusion models are effective for capturing multimodal action distributions, their iterative nature often restricts real-time control. To bridge this gap, the authors propose a spatiotemporal consistency prediction mechanism for high-quality "warm-start" actions and a velocity-aware perturbation strategy to prevent execution stalls. These methodological contributions are supported by a theoretical analysis proving the convergence of action errors during the refinement process. Empirically, the authors demonstrate that STEP significantly advances the Pareto frontier of latency and success rates. Most notably, with only 2 denoising steps, STEP achieves an average success rate 21.6% higher than BRIDGER and 27.5% higher than DDIM across the RoboMimic benchmark and real-world tasks. These results are validated through extensive testing on nine simulated benchmarks and two physical robotic platforms.

**Compliance With Llm Reviewing Policy:**

Affirmed.

**Key Questions For Authors:**

- The evaluation with a real robot is limited to two specific tasks. Could the authors provide further insight into the system's expected generalization and robustness when applied to more diverse or complex real-world manipulation scenarios?

 - Section 3.4 introduces a local contractivity analysis. Could the authors explain the practical implications of this theoretical analysis and specify if the underlying assumptions are expected to hold throughout the reported experimental evaluations?

**Limitations:**

yes

**Strengths And Weaknesses:**

Strengths:

- Principled Integration of Spatiotemporal Priors: The paper effectively addresses the "latency vs. quality" trade-off in diffusion policies by proposing a spatiotemporal consistency prediction mechanism. Unlike prior "spatial-only" (e.g., BRIDGER) or "temporal-only" (e.g., Falcon) methods, STEP provides a more holistic warm-start initialization that respects both the target action manifold and the continuity of robotic motion.

- Well-Structured Methodology: The paper presents a clearly organized technical approach in Section 3, differentiating between temporal, spatial, and spatiotemporal consistency. The integration of a predictor-based warm start with a velocity-aware perturbation mechanism forms a cohesive pipeline, which is documented through the schematics in Figure 2 and the procedural logic in Algorithm 1.

- Extensive Empirical Validation: The authors provide a comprehensive evaluation spanning several simulation suites and also tasks with a real single-arm robot. Data presented in Tables 2, 3, 4, and 6 shows the effectiveness of the STEP framework, demonstrating that it maintains high success rates and minimal latency.

Weaknesses:

- Theoretical Assumptions: The formal analysis in Section 3.4 is grounded in local Lipschitz and contraction assumptions. While these provide a conceptual framework for error convergence, the paper does not empirically verify these conditions, and there seems to be a gap between the high-level proof sketch and the practical performance of the system.

- Task-Dependent Performance: The superiority of the STEP framework is not uniform across all evaluated benchmarks. In specific cases, baselines like BRIDGER show competitive results, indicating that the relative advantages of the method may vary depending on the task dynamics.

- Scale of Physical Evaluation: The real-world robotic testing is conducted on a relatively small scale, involving two tasks with 20 demonstrations and 20 evaluation episodes each. While these experiments provide an initial proof of concept, the limited sample size and task variety offer constrained evidence for broader deployment claims.

---

> ### Author Rebuttal · Authors · 2026-03-27
>
> ### Q1: Theoretical Assumptions: The formal analysis in Section 3.4 is grounded in local Lipschitz and contraction assumptions. While these provide a conceptual framework for error convergence, the paper does not empirically verify these conditions, and there seems to be a gap between the high-level proof sketch and the practical performance of the system.
>
> We thank the reviewer for the valuable questions regarding our theoretical analysis in Section 3.4.
> We provide explicit theoretical and empirical justifications in https://anonymous.4open.science/r/STEP-1140/rebuttal/proof.pdf:
> Q1 and the second key questions both concern the **empirical validation of theoretical assumptions** and the **practical implications of local contractivity**, which we clarify and strengthen as follows:
>
> - **Lipschitz continuity validated**: We provide a self-contained theoretical proof that the diffusion policy is $L$-Lipschitz bounded, and empirically verify the denoising network’s Lipschitz continuity with explicit constants estimated via gradient norms.
> - **Contraction condition $c_k<1$ holds across all tasks**: We empirically confirm that the contraction coefficient satisfies $c_k <1$, validating that the contraction assumption holds in practice.
> - **Optimal warm‑start K predicted from measured MSE and verified experimentally**: From the empirical MSE of our predictor and our theoretical contraction condition, we determine the optimal warm-start step K for each task, which lies in the range 1–4 steps in Proof's section 2.2. This prediction is fully supported by the experimental results presented in the main paper.
> - **Practical implications**: The theory explains *why* warm-start speeds up convergence, identifies the optimal K, and guarantees stable error contraction. All theoretical assumptions are **validated empirically across all experiments**, closing the gap between theory and practice.
>
> ### Q2: Task-Dependent Performance: The superiority of the STEP framework is not uniform across all evaluated benchmarks. In specific cases, baselines like BRIDGER show competitive results, indicating that the relative advantages of the method may vary depending on the task dynamics.
>
> We acknowledge that performance variation across tasks is expected and aligns with theoretical predictions. The relative advantage of STEP depends on task characteristics:
> - **Task complexity**: STEP shows greater improvements on high-complexity tasks (ToolHang: **+700% over BRIDGER**) where spatiotemporal consistency is critical, while simpler tasks (Lift, Can) show smaller gaps as all methods approach ceiling performance.
> - **Multimodality requirements**: STEP preserves multimodal action distributions better than BRIDGER, which tends to mode-average in multi-solution scenarios.
>
> ### Q3: Scale of Physical Evaluation: The real-world robotic testing is conducted on a relatively small scale, involving two tasks with 20 demonstrations and 20 evaluation episodes each. While these experiments provide an initial proof of concept, the limited sample size and task variety offer constrained evidence for broader deployment claims.
>
> We sincerely thank the reviewer for this constructive suggestion regarding the evaluation scale. We agree that a robust physical evaluation is crucial for validating real-world deployment. To address this, we have significantly expanded our experimental campaign:
> - Increased Evaluation Episodes: We have increased the number of evaluation episodes from 20 to 100 per task. This larger sample size (200 episodes total across two tasks) provides a more rigorous statistical basis to evaluate the success rate and reliability of our method.
> |Method|PickNPlace|StackCube|
> |-|-|-|
> |DDIM-4step|0.95|0.81|
> |DDIM-2step|0.58|0.60|
> |Ours-4step|0.98|0.97|
> |Ours-2step|0.92|0.81|
>
> - Task Representative Power: While we focus on two primary tasks—pick-and-place and cube stacking—these are specifically chosen as they represent foundational challenges in robotic manipulation: **precise grasping, long-range transport, and multi-stage sequential assembly**. These tasks are widely accepted benchmarks in the robot learning community for evaluating sim-to-real transfer.
>
> We believe these additional experiments significantly strengthen the empirical evidence for our method’s robustness under real-world noise and closed-loop control.

---

> > ### Author Rebuttal · Reviewer_4LGY · 2026-04-04
> >
> > I am happy with the rebuttal. It answers my questions and concerns well. I keep my positive recommendation of this paper.

---

> > > ### Author Response · Authors · 2026-04-05
> > >
> > > We are very glad to know that your concerns have been resolved! We sincerely thank you for your careful reading and constructive suggestions. We will supplement the clarified explanations and details in the revised paper. Thank you again for your time and effort in reviewing our work.

---

### Decision · Program_Chairs · 2026-04-30

**Decision:**

Accept (regular)

**Comment:**

This is a technically solid and relevant contribution on accelerating diffusion-based visuomotor policies. The reviewers agree on the importance of the problem and the lightweight warm-start design, and the broad empirical evaluation in simulation and on real robots. The rebuttal strengthened the submission by adding theoretical clarification, more ablations, expanded real-world results, variance reporting, and code availability.

The main concern is that the novelty appears somewhat incremental, since the method combines existing spatial and temporal ideas rather than introducing a fundamentally new paradigm. The reviewers also raised questions about robustness, theoretical assumptions, and limited real-world validation, but most of these concerns were substantially addressed in the rebuttal. The paper appears to meet the acceptance bar, provided the final version clearly incorporates the rebuttal material and states its limitations more explicitly.